# SoftMatch: Comparing Scanpaths Using Combinatorial Spatio-Temporal Sequences with Fractal Curves

**DOI:** 10.3390/s22197438

**Published:** 2022-09-30

**Authors:** Robert Ahadizad Newport, Carlo Russo, Sidong Liu, Abdulla Al Suman, Antonio Di Ieva

**Affiliations:** 1Faculty of Medicine, Health and Human Sciences, Macquarie Medical School, Macquarie University, Balaclava Road, Sydney, NSW 2109, Australia; 2Computational NeuroSurgery (CNS) Lab, Macquarie Medical School, Macquarie University, Balaclava Road, Sydney, NSW 2109, Australia

**Keywords:** visual scanpath, Hilbert curve, discrete Fréchet distance, computational neuroscience, eye-tracking, fractal analysis

## Abstract

Recent studies matching eye gaze patterns with those of others contain research that is heavily reliant on string editing methods borrowed from early work in bioinformatics. Previous studies have shown string editing methods to be susceptible to false negative results when matching mutated genes or unordered regions of interest in scanpaths. Even as new methods have emerged for matching amino acids using novel combinatorial techniques, scanpath matching is still limited by a traditional collinear approach. This approach reduces the ability to discriminate between free viewing scanpaths of two people looking at the same stimulus due to the heavy weight placed on linearity. To overcome this limitation, we here introduce a new method called SoftMatch to compare pairs of scanpaths. SoftMatch diverges from traditional scanpath matching in two different ways: firstly, by preserving locality using fractal curves to reduce dimensionality from 2D Cartesian (x,y) coordinates into 1D (h) Hilbert distances, and secondly by taking a combinatorial approach to fixation matching using discrete Fréchet distance measurements between segments of scanpath fixation sequences. These matching “sequences of fixations over time” are a loose acronym for SoftMatch. Results indicate high degrees of statistical and substantive significance when scoring matches between scanpaths made during free-form viewing of unfamiliar stimuli. Applications of this method can be used to better understand bottom up perceptual processes extending to scanpath outlier detection, expertise analysis, pathological screening, and salience prediction.

## 1. Introduction

The maturation of eye tracking methods and bioinformatics over the last thirty years has led to novel methods used to sequence amino acids [1] for the study of genetics; and tracking eye movements for the study of human cognitive, neural, and perceptual processes [2]. The cross-pollination of DNA sequence matching algorithms with visual scanpaths, consisting of points where “objects of interest” are drawn to the fovea, has led to ScanMatch [3], a robust method for comparing visual scanpaths, and MultiMatch [4], a successor which uses the same bioinformatics algorithm but differs by matching scanpaths over many dimensions using geometric vectors. Both of these methods have been used and compared [5] in different applications, including detecting pathologies and characterising expertise and behaviour through gaze analysis. Gaze similarity was shown in experiments separating expert from novice participants while viewing brain MRIs [6], those separating healthy patients from those affected by autism [7], and during basic number search tasks [8].

A considerable amount of research within the last two years used ScanMatch and its vector-based implementation called MultiMatch to perform scanpath comparison [9,10,11,12], and both methods are featured in a 2021 paper on the state-of-the-art in human scanpath prediction by Kümmerer and Bethge [13]. Even though the research using both methods is varied and diverse, it relies heavily on a string editing matching algorithm developed over fifty years ago by Needleman and Wunsch [14]. Indeed, Needleman–Wunsch generated a lot of interest among researchers over a decade ago, including work done by Day [15] examining the validity of the Needleman–Wunsch algorithm in identifying and tracing the inner operations of cognition. His method, including those of ScanMatch and MultiMatch, involves two principal interdependent parts: a process-tracing step, followed by an analysis technique. Inevitably, fixation points need to be translated into 1D discrete representations. Translating areas of a stimulus in order to isolate ROIs into these representations has the added benefit of quantisation. However, this exposes a limitation. The simultaneous quantisation and isolation of these areas into string representations are done either through equally spaced boxed grids over a stimulus, or through unequally sized boundaries around specific areas of interest (AOI) to specify the “domains” of interest. Domains can be used when a small number of specific AOIs are being investigated, e.g., participants in a study with several buttons to choose from in a computer interface. Both the grid and domain dissection into AOIs incorrectly quantises points that are close to a boundary and also prevents locality preservation during string conversion.

This paper aims to build upon these bioinformatics-based methods with a new methodology called “SoftMatch”. The results show that in tasks where matching a gaze with a stimulus is required, our method performs better than others, even when the stimulus is both unfamiliar and highly complex, requiring the participant to use entry-level senses to process unintegrated sensory data, i.e., “bottom up processing”. Additionally, we have chosen to capture eye gaze data in a free viewing environment where the participant is asked to simply view the stimulus. This type of free viewing experiment has been found [16,17,18] to be a robust proxy for high cognitive function. This can be valuable in both measuring expertise and uncovering the underlying structure of perception. This paper hypothesises that string editing methods used to compare visual gaze patterns are best used in task based experiments, but are limited in a free viewing approach, where a combinatorial method to segment and measure sequences is more effective. This is accomplished by both implementing fractal curves, for increased quantisation performance, and incorporating a combinatorial discrete Fréchet distance calculation algorithm, which is sensitive to nuances between participants viewing a stimulus in no defined order. These stimuli could include any image, including medical images (e.g., MRIs), photographs, or abstract art. In this study, we choose to use creative paintings with participants instructed to view them freely. However, we do not make the assumption that viewers will examine the paintings in a similarly systematic way. Rather, our experiment aimed to define a similarity metric when the scanpaths compared do appear to be very different. We propose doing this by embracing a combinatorial method, departing from a string editing approach.

### 1.1. String Editing Methods

An introductory summary of the Levenshtein [19] distance metric provides context for the evolution of state-of-the-art scanpath and genome matching methods used today. It was the first string editing method used to match an ordered sequence to another. The distance represents a cost metric from a minimum execution of deletion, substitution, or insertion of characters within a string to make it match another.

However, a major shortcoming in using Levenshtein distance for gaze matching is in its inflexibility with both locality and time. Its dependence on fixed regions of interest (ROI) prevents granular discrimination between close and far points, and it does not account for differences in gaze duration. This was solved in 1970 with the introduction of the Needleman and Wunsch [14] sequence alignment algorithm. Similarly to Levenshtein, ScanMatch’s Needleman–Wunsch implementation uses string representations spanning an ROI grid over a stimulus. However, unlike Levenshtein, this method is able to find a best fit between two long strings by both allowing for gaps and also applying varying penalties when calculating substitutions during alignment. It does this by first creating a substitution matrix of all possible string combination scores; then a penalty is determined for gaps in the string; and finally, a score is added up, as an optimal path is calculated from the top left of the substitution matrix to the outermost column. Additionally, duration is implemented by repeating strings in a sequence proportional to others it surrounds. MultiMatch builds upon ScanMatch’s implementation of Needleman–Wunsch by examining multiple dimensions of a scanpath independently. While ScanMatch uses strings to represent gridded ROIs over the stimulus, MultiMatch uses strings to represent various quantised attributes of a participant’s gaze, such as its length, duration, and change in direction, to produce five different dimensions:Shape, used to measure the similarity in scanpath shape by producing the differences in aligned saccades as a vector.Length, used to measure the similarity in saccadic amplitude through the difference in saccade vector endpoints.Direction, used to measure the distance between saccades using their angles.Position, used to measure the Euclidean distance between aligned fixations.Duration, used to measure how long a fixation lingers between aligned fixations.

However, MultiMatch’s parsing and separation features can dilute statistical significance in highly complex and unfamiliar stimuli, as seen in this research’s results comparing gaze data from the paintings Cohen’s Blue Spot and Pollock’s Convergence, as seen in Figure 1. Furthermore, the quantisation used to isolate regions of interest into boxed grids over a stimulus in both methods prevents locality preservation during string conversion. These issues are described by Anderson et al. [5] as being “inherent in any measure using regions of interest or grids”. A review of human gaze will aid us in better understanding why Needleman–Wunsch algorithms struggle with such matches.

### 1.2. Human Gaze Physiology

Observation at its most basic level is the input of a visual scene as a whole, disseminated into details, with a re-assemblage of those details to form a combinatorial sensory percept [20]. The top-down neural decomposition of a visual scene coupled with the bottom-up building of details to form percepts in the brain is where combinatorial complexity exponentially increases. Assembling a synthetic model from this biological framework will give rise to rapid increases in dimensionality, causing an increase in the volume of space in the data. This large volume of space results in data sparsity, diluting the statistical significance within datasets. Having sparse data creates artefacts by obscuring similarities, preventing data organisation. Bellman coined this term in 1961 as “the curse of dimensionality” when considering problems in dynamic programming [21] and is especially debilitating in the application of machine learning within big datasets. For this reason, a handful of gaze modelling trends have emerged since Noton and Stark [22] first demonstrated that scanpaths may be replicated by the same viewer. This research opened a path for a large number of papers to tackle the challenge of clustering and measuring scanpaths [10,23,24].

In a 2020 analysis by Fahimi and Bruce [10], ScanMatch, MultiMatch, and other methods were compared in order to measure their discriminative power. The most contemporary method cited is by Anderson et al. [25]. In this research, recurrence quantification analysis (RQA), which is typically used to describe highly complex dynamic systems, is used to compare gaze patterns between participants cooperating in a study. Shortcomings in this study point to large differences between natural and gaze-contingent viewing, making accurate comparisons dependent on experimental parameters and participant behaviour. However, the inclusion of anisotropic visual behaviours in the returned results can provide a wealth of information about how gaze is affected by a stimulus, which string editing approaches do not provide. Recent work by Kumar et al. [23] attempts to address other shortcomings in string editing approaches using a weighted comparison matrix of pairwise comparison strengths, using various methods. These included Jaccard (JD) and bounding box (BB), longest common subsequence (LCS), Fréchet distance (FD), dynamic time warping (DTW), and edit distance (ED). All of these methods had strengths and weaknesses, but scaling the number and length of scanpaths presented a challenge, especially due to the lack of uniformity in results for each matrix clustering or reordering algorithm.

### 1.3. Saccades and Fixations in Scanpaths

A preliminary understanding of oculomotor behaviour will facilitate the interpretation of the many types of movements captured by a high resolution eye tracking device. These movements can include involuntary fixations, where an object of focus is kept in one’s visual field while scanning a scene. However, ocular fixation is not completely stationary. Involuntary physiological drift of the eye coupled with small perturbations at high frequencies often accompanies fixations, making the misleading implication that fixations are stationary [26,27]. Furthermore, the inhibition of return (IoR), which governs the frequency of attraction to fixation points, influences how long a gaze can be maintained. This makes drift, perturbations, and IoR critical behaviours affect the distribution of fixations during scene exploration [28].

Traditionally, fixations are stored in data structures using a similar method to how pixel positions are stored on a sensor array of a digital camera as (x,y) Cartesian points over a 2D plane. The challenge of both reducing dimensionality and increasing statistical significance for use in scanpath comparisons using string representations was achieved by creating a grid over the stimulus and assigning a letter combination to each square, as shown in Figure 2. Any fixation point with an (x,y) position would be reduced to a string name, which could then be compared to others using string editing methods such as Levenshtein or Needleman–Wunsch. For example, in Figure 2, the Cartesian points for the illustrated scanpath would be (2,1),(3,3),(4,2),(1,4), and its string equivalent would be BaCcDbAd. However, a major drawback to this method is both its crude quantisation calculation, where any point regardless of its proximity to a boundary is uniformly reduced to its grid value, and its lack of locality preservation, where once a fixation is reduced to its string value, the original precise position can no longer be determined. To address these issues, this paper proposes a scanpath representation model using fractal space filling curves.

### 1.4. Fractal Space Filling Curves

A fractal space filling curve is a theoretical line that travels through all the points in a space, in a self-similar fashion, without crossing (shown centre in Figure 2). The range of these curves could fill entire *n*-dimensional hypercubes of Euclidean space without endpoints. The continuous nature of fractal curves means it can fill a finite area as its perimeter wraps around its shape infinitely. Structures like this could be used as frameworks in machine learning algorithms such as *k*-nearest neighbour, where multidimensional points in a hypercube clustered on a space-filling curve can define a feature space [29]. For example, a Euclidean point converted to a 1D Hilbert curve can be graphed against time for a clearer empirical analysis than if it were left in 2D space with an additional third temporal dimension. Furthermore, normalisation of multidimensional data in a 1D space would lead to increased precision as pixel resolution increases, without the need for a linear piecewise function used to interpret grid changes used in string editing methods. This is because the curve preserves locality well due to its homeomorphic exponential growth with the *n*th approximation of the limiting curve. Thus, as it increases in size, its ratio of detail to scale remains relatively constant. This example of the Hilbert curve’s regularity of self-similarity is a testament of its robust ability to preserve locality during a shape’s growth or change during variable quantisation [30]. This makes it particularly robust in applications where 2D values are reduced to 1D fractal curves plotted against a time dimension, to be both dynamically quantised over space and uniformly windowed over time.

### 1.5. Recurrence Measurement with Multidimensional Data

Reduction in dimensionality can be particularly useful when comparing two dimensional locations to one another. Indeed, this is critical in string editing methods where grid substitutions can turn a sequence of 2D values into a 1D string sequence. These 1D values can be grouped into a subset of a metric space and compared with others to determine a distance metric. For example, the Hausdorff distance measures the maximum distance between all closest rival points in two sets as the overall distance metric between them. Therefore, an unordered set of 1D Hilbert distances could be the members of a set matched to another where the closest points are iterated for maximum distance. Adding an additional dimension for time would allow for a more detailed context and therefore, better representation of fixation points in a scanpath during comparison using only two dimensions: (h,t), where *h* = Hilbert distance and *t* = time.

A robust method for the measurement, prediction, and analysis of patterns in nature can be found in the work done by Webber and Zbilut [31] describing methods for recurrence quantification analysis (RQA). The theoretical premise behind the RQA of natural patterns is that a direct relationship can be found connecting recurrent patterns and their underlying dynamics. A simple example introduced by Webber and Zbilut to describe RQA used wave heights measured by a buoy bobbing on ocean waves. In a plot of wave heights measured against time, a demarcation is placed at one chosen height, e.g., 0.9 ft, and the time points of all waves are measured at that height. A new plot of 0.9 ft points is made to measure the frequency of those recurrent points in time. Incorporating other wave heights will show the distribution of the 0.9 ft height in comparison to all other heights in a plot measuring the comparative recurrence of those heights by their corresponding times. This concept is applied to eye paths by Gandomkar et al. [32] in order to distinguish expert radiologists from less experienced ones examining mammographic images. The authors introduced RQA in order to address spatio-temporal dynamics that are absent in time-related metrics, e.g., fixation latency, viewing times, target fixation duration, total fixations, fixations cluster sizes, and fractal dimension [32]. Unlike these approaches, RQA considers scanpaths of complex sequences; fixation points and their corresponding time values can be recurrent within a scanpath. Similarly to the wave example illustrated earlier, fixation points are quantised to a 2.5° radius of a previous fixation and are plotted using eight metrics to evaluate their positions in space over time. Three examples of these metrics include recurrence (REC), defined as the percentage of all fixation pair combinations that are quantised to the same position, $T_{2}$ as the difference in average time between two non-consecutive returns, and laminarity (LAM), which is the measure of a set of consecutive fixations repeated many times in a scanpath. Using four experienced and four inexperienced radiologists viewing 120 mammograms, Gandomkar et al. were able to reveal that experienced radiologists were more efficient in their deterministic, laminar, and re-fixating eye movements.

However, unlike the Hausdorff and RQA methods, our method incorporating discrete Fréchet distance calculations can fully exploit the additional time dimension. A common allegory used to describe how Fréchet works is that of a man and a dog, both walking in a single direction down their own different paths [33], where the length of the leash is the smallest of the maximum pairwise distances necessary for the two to remain connected as they both stop at each vertex towards the end of the path while travelling in the same direction. For example, Figure 3 illustrates how the same points can return a small distance metric when measured without an ordered sequence of points using Hausdorff (Figure 3, left) versus a much longer distance metric when measured with the actual order of points using discrete Fréchet distance (Figure 3, right). Indeed, a singular approach using aligned and ordered items in a collinear comparison methodology would be prone to problems. Artefacts can be produced when comparisons are weighted too heavily on conforming to an ordered sequence, as can be encountered in bioinformatics, where DNA is prone to mutagens, and in visual scanpaths, where anisotropic effects can influence attention.

### 1.6. Problems with Assumed Collinearity

In the field of bioinformatic sequence matching, where Levenshtein and Needleman–Wunsch are heavily used, there is some debate regarding aligned verses unaligned sequence matching. Zielezinski et al. [1] identified five cases where alignment-based methods would introduce problems with amino acid matching results, all of which translate to similar issues using Needleman–Wunsch implementations in scanpath matching tools such as ScanMatch and MultiMatch. By examining parallels in both fields, some prescient insight can be made to mitigate similar limitations in scanpath analysis. First, both in DNA sequencing and scanpath analysis, aligned comparison methods depend on matched collinearity, i.e., the homologous sequence of conserved linear visual fixations (in neuroscience) or amino acids arrangements (in bioinformatics). Indeed, this assumption is demonstrated in a ScanMatch tutorial where the matched scanpath data is composed of participants following a numerical visual track task [34]. In reality, both scanpaths and genomes do not follow such uniform arrangements; scanpaths are combinatorial in nature, and genomes possess a high degree of variation due to increased rates of mutation [1]. Second, random sequences can mix with remote homologs when the identity, or in the case of ScanMatch, the substitution matrix, contains too few values. This can be further exacerbated when gaps are allowed [1]. Third, the memory requirements for creating all possible sequences of either a genome or scanpath in a substitution matrix scales exponentially with length. Fourth, as just mentioned, the rapid scaling of long sequence alignments quickly approaches an NP-hard state where solving a match quickly becomes intractable. This results in shortcuts to optimise matches that may introduce artefacts [35]. An example of the introduction of such risks could be demonstrated in a recent attempt at using crowd-sourcing for sequencing DNA with the application Phylo [36]. Finally, the parameters and matrices used to corral both amino acids and scanpaths into a tenuous match offer a scoring system that is not shared between alternate applications or methods. Even within its own method, slight variations in parameters can provide substantially differing alignments. As these sequence alignment methods rely on a priori mappings of amino acid and fixation sequences, they both betray the combinatorial structure of scanpaths and also demand empirical fiddling with arbitrary parameters.

A solution to all these issues would be to separate sections of a sequence into equally sized portions and compare them with other portions in an aligned manner, allowing for the combinatorial nature of the data to facilitate matches. This research proposes that scanpath sequences of fixations over time (loose acronym SOFT) can be used to match with others to move beyond the rigid definition of collinear aligned sequence matching and move towards a more physiologically representative combinatorial model. Put more simply, participants are not penalised for viewing regions of interest at different times within the scanpath during comparison.

## 2. Methods

Figure 1 reveals the six stimuli used in this experiment: Jackson Pollock’s *Pasiphae* (1943), *Convergence* (1952), and *Blue Poles* (1952) capture high degrees of abstract complexity. This is polarised by Bernard Cohen’s *Blue Spot* (1966) painting, which may match Convergence in emotional impact, but differs by displaying a lower degree of geometric complexity. Vincent van Gogh’s familiar *Starry Night* (1889) painting provides a vibrant contrast to William Turner’s *The Slave Ship* (1840). Participants were not guided to view anything in particular and had no known art training or critical instruction. This paper proposes a similar approach for scanpath comparison as other binary comparison methods such as ScanMatch and MultiMatch, where the 2D scanpath is reduced in dimensionality before implementing their respective matching methods. However, our proposed method uses Hilbert distances instead of boxed grids for dimension reduction. As a result of not using grids, quantisation is done in the comparison phase in lieu of integration into the preprocessing box-gridding phase, as is done in other methods, e.g., ScanMatch and MultiMatch. This will both preserve locality and decouple quantisation from the dimensionality reduction process.

The benefits of using fractal curves can be understood when comparing it to string editing quantisation, which uses boxes to designate regions as letters. The gridded boxes are used as both a quantisation method to snap all fixations within a box to a single value, and also as a way to designate 2D coordinates as a 1D string. Once a coordinate is assigned to a box with a designated letter, it is unable to be converted back to a 2D coordinate without information loss due to the quantisation during dimensionality reduction to a string value. Fractal curves mitigate these issues because the curves pass through all points in a space, which means a 2D point can be converted to a 1D point, and back, without loss. This also means a point on a fractal curve can be quantised independently of its dimensionality reduction, unlike box methods, allowing for more flexibility when trying to find optimum values, which could be exploited using future machine-learning-based optimisation methods.

However, Hilbert distances alone will suffer the same disadvantages as gridded string substitutions if time values are ignored. By adding time values as a dimension, a 2D Hilbert versus time axis can be constructed. This opens up a large number of distance comparison metrics between two sets. This paper diverges from other scanpath comparison methods by comparing combinatorial sequences of fixations over time, diminishing the significance of the order by which people examine a stimulus. Instead of a long, single, sequential list of fixations making up the baseline for comparison, the scanpath will be cut into short sequences of fixations, in equal time window lengths, which we call tau (τ). Figure 4 illustrates the preprocessing stage where Cartesian fixations are appended with their Hilbert distances before Step 1 segments a scanpath into equally sized time window bins using parameter τ. Step 2 measures the discrete Frechet distance between two segments, and if it is <δ, adds +1 to the cumulative score. Step 3 compares all scores in a group; lower scores indicate less similarity.

### 2.1. Stimuli and Participants

Experiments were conducted by a trained researcher and approved through The Faculty Ethics Subcommittees at Macquarie University in accordance with the Australian National Statement on Ethical Conduct in Human Research. The 53 healthy participants, labelled P01 through to P53, included medical professionals who were enrolled in a broader eye-tracking machine vision study in which medical and non-medical images (e.g., paintings in this study) were used. Exposure to stimulus was preceded and followed by exposure to noise. Participants were asked to examine multiple images, including the digital reproductions of six artworks illustrated in Figure 1. Their gaze was captured by the eye-tracker EyeLink^®^ 1000 Plus (SR Research, Ottawa, ON, Canada) operating at 1000 Hz at 0.05° root-mean square (RMS) and 0.25° saccade resolution. Both eyes were captured during tracking. However, only one eye was used for computation to maximise tracking accuracy. This decision was partly made due to work by Hooge et al. [38] in their paper “Gaze tracking accuracy in humans: One eye is sometimes better than two”, which demonstrated that one eye measurement can reduce systematic error in computed measurements. Additionally, no part of this experiment was reliant on binocular dynamics, further strengthening the case for a single eye measurement. The raw samples used directly from of the eye tracker can be found in the accompanying data labelled “Preprocessed Data”. The head mounting was free to move; fixations from both eyes were saved into a matrix consisting of the trial number, participant ID, eye fixations, saccades, blinks, and a timestamp for each captured event. To reduce unwanted data, post-processing was used to reduce the data to four columns representing: right eye (x,y) coordinates, its position converted to a Hilbert distance, and a duration for each fixation.

### 2.2. Fixation Position Using Hilbert Curves

This paper aims to provide an alternative to 2D quantised gridding by using a space-filling one dimensional fractal curve. A 2D point on a 1D fractal space-filling curve both reduces the complexity of a measurement by using only one dimension, and also preserves locality when it is converted back to 2D space. This research proposes that a 1D coordinate representation of points in the shape of the Hilbert curve is better suited to represent a scanpath’s position. It can perform better than string representation because quantisation is separated from dimensionality reduction when using fractal curves. This provides the ability for fractal curves to optimise quantisation while in 1D space.

### 2.3. Outlier Identification

Oddball scanpaths with valid fixation points and saccades, all within the boundary of the stimulus, but lacking features shared by the majority, could either represent a blunder in the data collection process or could reflect a critical yet isolated aspect of the scanpath. Surprisingly, a formal definition of outlier scanpaths does not exist, even as researchers computationally strive to cluster and compare entire scanpath datasets (Burch et al. [39], Jolliffe [40]). In such cases, a judgement call must be made: Does this scanpath represent a software glitch, an artefact of the experiment process, inattention and distraction, or a lack of expertise? Alternatively, does the scanpath represent a valid edge case which would greatly influence data boundaries during clustering? Indeed, robust analysis keeps data that are unusual and significant while removing artefacts.

In research by Newport et al. [37], these ambiguities were mitigated by a more detailed geometric complexity metric achieved via the fractal dimension. This is done firstly by fitting the 2D scanpath into a Hilbert curve and then measuring it as a sequence of fixations using the Higuchi fractal dimension (HFD). The outliers from HFD analysis are then compared to non-matching results with scanpath matching tools (e.g., SoftMatch) to robustly identify defective data. These results will be highlighted in a clustered heatmap, which is a graphical representation of data frequently used in bioinformatics to illustrate clusters in hierarchical matrices of data. In this research, these methods were used to find outliers which could have also been present in clustered heatmaps made by SoftMatch, and was evaluated for exclusion.

### 2.4. Time Binning

The purpose of data preprocessing, shown in Figure 4, is to reduce the dimensionality of the Cartesian coordinates and a assign duration to each point in order to construct spatio-temporal tuples in a 2D scanpath array. This array can then be used in Steps 2 and 3 in Figure 4 when assigning parameters τ and δ. This will result in each scanpath fixation containing a matching duration value representing the amount of time the participant has spent gazing at that specific fixation position. A SoftMatch sequence vector is an ordered list of location and duration pairs separated in a uniform time window (τ) which is *N* milliseconds in size. These vectors are all contained within the parent scanpath and each are compared, one by one, to other scanpath collections of vectors in a combinatorial fashion. Figure 5 illustrates how fixation points converted into Hilbert distances can be used as a virtual axis in an imaginary Hilbert versus duration space. In this example, the tau τ window carves up these pairs of (h,d) points into 6 s bins. If a duration contains a remaining number of milliseconds when binned, that (h,d) position is repeated with the remaining portion used when summing durations in the next Soft segment.

An empirical approach to picking the size of time bin τ, which is used to separate a scanpath into many equally sized SoftMatch segment vectors, can be undertaken by manually averaging periods of time spent by the participant between empirically defined regions of interest in a stimulus. When estimating parameters empirically, longer time bin windows (τ) can include too much fixation sequence detail, making matches more difficult, whereas shorter τ values will lack enough detail to make matches statistically and substantively significant.

Though it is outside the scope of this paper, an approach using the brain’s own rhythmic attention physiology could provide a baseline value for time bin window parameter τ. An “attention window” size of approximately 8 Hz, or 0.125 s per period, was defined by research performed by Nakayama and Motoyoshi [41]. This research illustrates that attention can bind visual features as single events in a chain of perception. This fits well with the definition of τ defining a fixed window of time used to carve a scanpath into sequences of equally sized fixation segments. Establishing a clinical trial testing the neurological best fit for brain cycles using this method is outside the scope of this paper. However, replacing mathematically derived best estimates with “attention windows” of 8 Hz or 0.125 s per period, similarly to the phase-locked neural oscillations described by Nakayama and Motoyoshi, may provide a good starting point when empirically exploring parameter values during the development of computational biological models involving attention. Indeed, SoftMatch uses this attention window as a default value for τ=90 ms.

### 2.5. Measuring Curve Similarity

As this study will often be comparing one set of points to another, each point in each set will be compared to a point in a different set described as its adversary. As the points in each set are compared, they will be termed adversarial pairs to describe both their status as points in different sets, and also as points within a comparison metric. The measure of similarity between two points can be measured as the distance *d* between both their *x* and *y* coordinates. However, the measure of closeness between two point sets does not address the sequential and temporal aspects of a scanpath. In our method, we propose reducing a coordinate’s 2D *x* and *y* to a singular Hilbert distance *h*. In addition to the *h* axis, we add a temporal axis *t*. This is to create a reduction of 3D space-time (x,y,t) into a 2D spatio-temporal axis (h,t), for each of the scanpath fixations and corresponding times. Therefore, the distance between the Hilbert *h* time axis *t* is measured between points as $d=\sqrt{{\left(h_{2}-h_{1}\right)}^{2}+{\left(t_{2}-t_{1}\right)}^{2}}$. However, with multiple points and a variety of distances comes additional complexity when attempting to measure similarity. This method uses discrete Frechet distance to analyse recurrence using the closeness between two curves. The original Fréchet distance formula measures distances from all possible points along the curve. However, the *discrete* variant restricts measurement of distances to specific “discrete” vertices along its curve, rather than any and all points. This suits our method because fixation points represent vertices on our polygonal scanpath curves. The following equation mathematically represents how the discrete Fréchet distance $d_{F}\left(A,B\right)$ performs:(1)$$\begin{array}{cc} \; & \mathrm{Let}\;M\;\mathrm{be}\;a\;\mathrm{metric}\;\mathrm{space}.\\ \; & \mathrm{Let}\;\mathrm{curve}\;A\;\mathrm{and}\;B\;\mathrm{be}\;\mathrm{two}\;\mathrm{non-empty}\;\mathrm{subsets}\;\mathrm{of}\;a\;\mathrm{metric}\;\mathrm{space}.\\ \; & \mathrm{Let}\;d\;\mathrm{denote}\;\mathrm{the}\;\mathrm{distance}\;\mathrm{function}\;\mathrm{of}\;M.\\ \; & d_{F}\left(A,B\right)=\underset{\alpha ,\beta }{\inf}\;\max_{t\in [0,1]]}\;\left\{d\left(A(\alpha (t)),B(\beta (t))\right)\right\} \end{array}$$
When making *t* an informal representation of time, A(α(t)) and B(β(t)) represent adversarial points at any given time *t*. Requiring increasing α,β movement from its greatest lower bound (i.e., through its infimum) encourages forward movement along the curve. The infimum over all re-parametrisations of [0, 1] describes the minimising distances between consecutive adversarial points while progressively iterating along the curve. The final result of $d_{F}\left(A,B\right)$ is a singular distance metric between curves *A* and *B*. When the distance metric for two SoftMatch segment vectors is returned, it is compared against the quantisation parameter δ, which represents the maximum distance for a match between two curves. The final function of the sum of all segments, or SoftMatch function, can be described as such:(2)$$\begin{array}{cc} \; & \mathrm{Let}\;i,j...n\;\mathrm{represent}\;\mathrm{each}\;\mathrm{fixation}\;\mathrm{sequence},\\ \; & D\;\mathrm{the}\;\mathrm{distance}\;\mathrm{returned}\;\mathrm{by}\;d_{F}.\\ \; & \sum_{i,j=1}^{n}d_{F}\left(A_{i...n},B_{j...n}\right)=\left\{\begin{array}{cc} 1 & \mathrm{if}\;D<\delta ;\\ 0 & \mathrm{otherwise}. \end{array}\right. \end{array}$$
If a match is determined, the match score for the pair is incremented by one point. No match provides no points. After all the SoftMatch segment vectors in one scanpath are compared to all those in another, a final score is returned, determining the overall match score value for the pair. No normalisation is done in order to introduce interpolated points into each curve. When two “curves” are compared, they are comprised of points which are used as the discrete vertices of a curve, as shown by the illustration on the right in Figure 3. The way that each curve is “portioned” equally is through time windowing using parameter τ, introduced in the next section.

### 2.6. Method Parameters

This method incorporates two parameters used to create a SoftMatch segment vector. The first (τ) determines the size of the SoftMatch time bins, in order to capture greater or fewer fixations. The second (δ) is used to quantise fixation location data, in order to increase their statistical significance during the SoftMatch matching process.

#### 2.6.1. Quantisation

The method outlined in this paper quantises fixation locations by implementing a parameter, denoted as δ, representing a maximum distance two curves can be from each other to match. Figure 6 illustrates this, where the grey circle outlines the boundary for inclusion of the maximum discrete Fréchet distance between the fourth point and its adversary. The tolerance threshold discriminates against unmatched curves by establishing a maximum discrete Fréchet distance for matches. In contrast, grid-based quantisation methods such as ScanMatch and MultiMatch require that all fixations falling inside a grid square assigned the same location attribute when compared to each other. In Figure 6, the two scanpaths on the left have fixation points that lie very close to each other in Cartesian space, yet are quantised to be further apart due to their positions being close to the grid boundary. The string edit representations of the scanpaths in Figure 6 would be AbBdDaCc and AcBcCaDc, which are almost completely different. Alternatively, the 1D Hilbert representations would be (17,8,41,56) and (16,9,40,52), yielding a more accurate and quantifiable representation of their similarity. Adding a temporal dimension will create 2D Hilbert–duration curves. SoftMatch uses a discrete Fréchet distance to measure between adversarial curves; a maximum distance is required to return a match, indicated via a tolerance parameter δ in Figure 6. The measurements are made originating from each point’s spatio-temporal position. This mitigates quantisation issues in grid-based methods such as ScanMatch and MultiMatch, where adversarial fixation points, which may fall close to each other on a grid boundary, are separated due to their positions in different gridded squares.

#### 2.6.2. Time Binning

The binning window, shown in Figure 7 and denoted as τ, is the second and final parameter used in this method. It defines the length of time per span as a uniform number of milliseconds per segment. The time bins must be kept at a defined length because this method is based on the frequency of subsequences within it. As described in Section 2.3, a good starting point for this parameter could use the brain’s rhythmic attention network, as defined by Nakayama and Motoyoshi [41] to be 8 Hz, or 0.125 s. Future work is planned to use an optimisation analysis method using machine learning to find the best fit for τ and δ. Nevertheless, SoftMatch provides the flexibility to use any other type of optimisation method in order to optimise τ and δ granularity.

## 3. Statistics and Testing

The metric of success for the methods tested in this paper is the magnitude of the difference between concordant and discordant groups of scores. One method of analysis is not enough [42] for measuring differences between groups, since *p*-values are suited only to statistical significance, whereas an effect size provides a substantive significance. Therefore, a combination of heatmaps, Cohen’s effect size, and paired *t*-test *p*-values were used to measure how effective SoftMatch, ScanMatch, and MultiMatch performed in binary tests. Tests were composed of scores from concordant (A versus A and B versus B) and discordant (A versus B) matches. Paired *t*-tests calculated *p*-values from two randomly picked (without replacement) sets of 100 match scores from each comparison, which we repeated 1000 times and averaged. Picks were chosen from the triangular wedges shown in Figure 8, which included 1378 unique participant combinations. Cohen’s effect size was computed using all 1378 unique match combinations. Heatmaps are included to provide an empirical validation of *t*-test and Cohen’s effect size results.

The clustered heatmap supports *t*-test results measuring the *p*-values between identical and discordant match results, shown in Table 1. Paired *t*-testing was conducted with a noted dependency between the same participants used when scoring matches between identical (e.g., Convergence vs. Convergence) and discordant (e.g., Convergence vs. Blue Spot) stimuli. Each sample used in the paired *t*-test included random pairs of participants; no single participant was used more than once per test, giving non-repeating pairs, as shown in Figure 8. The participant sample’s identical match scores, e.g., Convergence vs. Convergence and Blue Spot vs. Blue Spot, were tested against the participant sample’s discordant match scores, e.g., Convergence vs. Blue Spot. This was done to see if one discordant group was more significantly separable from one concordant group than another, e.g., “Convergence versus Blue Spot” scores being more significantly separable from “Blue Spot versus Blue Spot” than “Convergence versus Convergence”.

## 4. Results

SoftMatch was tested using two different approaches. Firstly, artificial scanpaths were created, and matching was scored based on comparisons to a noise-perturbed duplicates. This experiment was adopted by Dewhurst et al. [4] to test their MultiMatch method against ScanMatch, and Cristino et al. [3] to test their ScanMatch method against Levenshtein. In this research, we continued developing this experimental framework by adopting the same artificial scanpath experiment with our method against MultiMatch and ScanMatch. To be clear, we adopted this approach using synthetic scanpaths because it was used by both ScanMatch and MultiMatch to show how effective each method is with controlled perturbations augmented in each synthetic scanpath sample. Secondly, real world scanpaths were compared using 53 participants viewing six different paintings exhibiting varying levels of abstraction (as shown in Figure 1). Accuracy was determined by comparing match scores; higher scores mean greater matchability. Our hypothesis proposes that our combinatorial method will return higher scores when matching the gaze patterns of different participants looking at the same stimulus. For example, we propose that gaze patterns looking at William Turner’s *The Slave Ship* will match better with other gaze patterns looking at the same thing, versus those of Vincent van Gogh’s *Starry Night*. This was tested using heatmaps, Cohen’s effect size, and *p*-value scores to see if there is a difference between each score of discordant and concordant matches.

### 4.1. Artificial Scanpath Matching Experiment

Three synthetic scanpaths were generated in order to estimate and compare the sensitivity of SoftMatch to artificial noise: S1, S2, and S1p. S1 and S2 scanpaths were populated with 10 randomly generated sequential fixation positions. S1p was a copy of S1 perturbed with noise from a Gaussian distribution. The noise varied with a standard deviation (σ) ranging between 10 and 90% of the screen width (W). Duration was a random number of milliseconds of between 150 and 300 ms per fixation, representing average fixation duration [43]. Duration was perturbed with noise from a Gaussian distribution ranging between 10 and 90% of the difference between 150 and 300 ms. Figure 9 illustrates a random set of S1, S2, and S1p scanpaths plotted in 2D space.

The experiment included five levels of perturbation, including (σ) = 0.1, 0.3, 0.5, 0.7, and 0.9 of *W* and 24 unique (S1, S2) pairs. Each level or perturbation was applied 50 times to each unique S1 scanpath to create an adversarial S1p value, making (S1, S2, S1p). This created a total number of 1200 samples per (σ) perturbation (24 unique scanpath pairs multiplied by 50 perturbations per σ), adding up to 6000 samples in total for SoftMatch, ScanMatch, and each MultiMatch attribute. The match method correctly classified the perturbed scanpath if the comparison score was lower between S1 and S2 than between S1 and S1p.

SoftMatch was assigned (δ) = 0.0 in this experiment due to the low number of fixation points (10 samples), making quantisation unnecessary. The maximum value of all 10 fixation durations (max(d)) was used to calculate τ. ScanMatch, MultiMatch vector, direction, length, position, and duration results were included using the default settings in the Matlab MultiMatch toolbox. It should be noted that this experiment included MultiMatch duration, which was omitted in the original Dewhurst et al. [4] experiment.

Figure 10 illustrates SoftMatch results against MultiMatch and ScanMatch. Indeed, it appears that in small perturbation amounts, SoftMatch did not perform comparatively well with MultiMatch, especially in direction and position. ScanMatch performed better than SoftMatch with less noise but performed equally or slightly worse in high noise tests. However, as spatio-temporal noise was increased, SoftMatch did appear to improve over MultiMatch in duration, approach length, or vector performance. The reason MultiMatch’s direction and position were good may be because S1p was not perturbed sequentially, allowing lower sensitivity attributes such as position and direction to isolate themselves from higher sensitivity attributes such as vector, length, and duration. This kind of isolation may also be a weakness for MultiMatch, since researchers are left to draw their own conclusions from five potentially very divergent MultiMatch feature results, as shown in Figure 10. Furthermore, this experiment reinforced scanpath collinearity by maintaining the spatio-temporal order of all fixations regardless of perturbation amount. In a task-based, sequential experiment where participants are rewarded for pursuing a particular order, this type of experiment would work well. However, in a free-viewing experiment designed to be a proxy for high cognitive function, where there is no task, perturbations would include disturbances to the sequence order of fixations, in addition to spatial noise, exposing a weakness in these methods for matching non-sequential scanpaths. In the following experiment, we will see how a free viewing experiment will reveal the limitations of ScanMatch’s and MultiMatch’s reliance on spatial colinearity.

### 4.2. Real Scanpath Matching Experiment

To measure the accuracy of our method with real scanpaths, we used an existing dataset (described in Section 2.1) of 53 eye tracking trials where each person looked at a painting without any particular task or instruction. Participants were medical professionals with no formal art training. After all participant scanpaths were recorded, a score was calculated using SoftMatch, ScanMatch, and MultiMatch by comparing one participant’s scanpath with another. In some cases, the paintings were the same, or concordant, and in other cases they were discordant, or different. The list of scores for when the painting was concordant was compared to the list of scores where they were discordant, to determine if the scores were different enough to be (*p*-value <0.05) significantly so. Success was defined as scores for concordant results being consistently higher than discordant ones. Default values were used with SoftMatch, ScanMatch, and MultiMatch.

A clustered heatmap, used in bioinformatics to illustrate clusters in hierarchical matrices, was used to illustrate this point by revealing lower comparison scores between participants who looked at Stimulus A versus participants who looked at Stimulus B (see Figure 8). This was done with each axis representing all participant stimuli combinations along each axis to empirically reveal separability between the groups. *p*-values (<0.05) were calculated to determine the statistical significance of separation between concordant and discordant scores. We conducted 1000 paired *t*-test trials using 100 randomly picked (without replacement) scores to calculate the *p*-value. Cohen’s effect size was used to determine the substantive significance (effect size >0.20) and confirm observations seen in the heatmaps.

An outlier evaluation was done using Newport et al. [37] Higuchi fractal dimension (HFD) analysis, as shown in Figure 11. The results on the y axis are geometric complexity values, measured via a scanpath’s HFD. Horizontal lines represent standard deviations from the mean for all participants viewing the stimulus. A scanpath was deemed a potential outlier when its fractal dimension is outside two standard deviations from the mean of all others in a stimulus group. Figure 11 shows potential outliers at Blue Spot P47 and Convergence P14 and P51. These scanpaths were inspected during SoftMatch scoring as potential outliers. Other corroborating anomalous results returned from these three scanpaths during matching provide a robust justification for their exclusion from further study. No outliers required removal, and thus all participant scores were used in *p*-value results, Cohen’s effect size results, and heatmaps which can be found in the Appendix A.

#### Reliability and Uniformity

The clustered heatmap seen for SoftMatch in Figure 12 illustrates a high degree of visual separability in matching results for both pale low scores and darker high ones. A decreased magnitude of difference for un-matched scanpaths (e.g., Convergence and Blue Spot) was to be expected, since matching stimulus scores may contain greater variance (e.g., 154, 115, 87…) compared to unmatched stimulus scores which are always close to zero (e.g., 5, 8, 0…). As shown in Table 1, SoftMatch returned the highest number of comparisons that were statistically significant—24 out of 30 returned a *p*-value less than 0.05; and the highest number of comparisons that were substantively significant—27 out of 30 returned a Cohen’s effect size greater than 0.20. The effect sizes can be visually validated by noticing the chequered pattern illustrated in the complete list of heatmaps found in the Appendix A.

ScanMatch also performed well, but was slightly behind SoftMatch in statistical and substantive significance, falling two samples behind in both. A view of the detailed breakdowns (please see Appendix A) by ScanMatch and SoftMatch, there was a shared difficulty in establishing the magnitudes of differences between concordant and discordant scores for Convergence versus Blue Poles and Convergence versus Starry Night. ScanMatch lost to SoftMatch in tests of Blue Poles versus Pasiphae, Convergence versus Pasiphae, and Convergence versus The Slave Ship. However, ScanMatch did manage a better *p*-value against SoftMatch when comparing Starry Night versus The Slave Ship, though SoftMatch missed the 0.05 cutoff here by 0.0024. All results of the analysis can be found in the Appendix A.

Conversely, the clustered heatmap for MultiMatch seen in Figure 12 demonstrates the difficulty of comparing natural scanpaths in a free-form, unstructured viewing experiment. This is consistent with other stimuli, and a complete set of heatmaps can be found in the Appendix A. Multiple participant matches appear to have no MultiMatch scores at all, which may be attributed to the nonlinear and unstructured nature of the viewing experiment. Indeed, Table 1 illustrates how poorly MultiMatch results did compared to SoftMatch and ScanMatch in both *p*-value and Cohen’s effect size. MultiMatch vector and direction did better than other MultiMatch results, which may have been due to erratic length, position, and direction behaviours.

A look at HFD measurements in Figure 11 indicates that there is no correlation between scanpaths which match poorly in the MultiMatch results shown in Figure 12 and those with geometric complexities outside two standard deviations from the mean. However, SoftMatch was able to find Convergence P14 displayed in Figure 12 as a white line, meaning it is a very poorly matching scanpath, but did not detect Blue Spot P47 or Convergence P51. Corroborating evidence of outlier status using the different approaches in SoftMatch and HFD outlier detection by Newport et al. [37] may provide a solid basis for exclusion of Convergence P14 from further study.

## 5. Discussion

The results indicate that a combinatorial approach, also used in amino acid matching, can produce improvements with scanpaths. By using six paintings as ground truth, a clear distinction can be made between what should be a high-scoring match between identical paintings and a low-scoring match between two different ones. The high- and low-scoring match results are shown to be statistically distinct during a rigorous *t*-test and returned robust Cohen’s effect sizes, indicating scanpath matching had statistical (*p*-value >0.05) and substantive (>0.20) significance, as shown in Table 1.

The implementation of Hilbert distances instead of traditional 2D grids is instrumental in the increased precision of quantisation during matching. This is due to both a reduction of errors where two points are on either side of a grid boundary, causing over-quantisation, and also increased flexibility for optimisation, where quantisation can be adjusted during method execution, without the need for pre-processing with, e.g., gridded methods. However, we did not provide a formal ablation study where we replaced our Hilbert and time (h,t) distance metric with other Euclidean (x,y) or alternatively quantised (q(x),q(y)) methods because we chose to focus on its comparison to string editing techniques. An alternative method to discrete Frechet distance measurement using weighted and optimised τ and δ values could provide better results. For example, using a support vector machine (SVM), the SOFT segments can be implemented as multidimensional feature spaces for each stimulus observation. Training would fit a margin around these features, separating it from others during a binary classification task. However, this is reserved for future work, where this combinatorial approach could provide a baseline for more advanced machine learning based approaches.

SoftMatch’s conversion of scanpaths into axes of Hilbert distance over a time dimension would, at first glance, appear to be a good match with RQA analysis. It uses time to detect unstable periodic orbits and transitions between repeating clusters and can create cross recurrence plots to discover similarities between processes. However, the clue to its limitation with scanpaths lies in an example Webber [31] used describing RQA ECG signal analysis. The signal, being a periodic 1D representation of voltage as a function of time, has a regular and consistently measured time dimension. Comparing this to scanpaths yields a similarity where eye movements are captured over time as fixation points, with the important difference that fixations are inconsistently spaced along the time dimension; some fixations linger for very long or very short periods. This means that implementing RQA requires the interpolation of fixation points which will diminish the statistical significance of lingering fixation points and fleeting glances that may yield important clues when comparing participants’ gaze sequences. For this reason, the method used by this paper uses a curve to describe each windowed scanpath segment, and the distance between two curved segments as a measure of its similarity.

The poor performance of SoftMatch in the simulated scanpath experiment, used by both ScanMatch and MultiMatch to demonstrate their accuracy, exposes serious limitations. However, the question of whether the problems are inherent in the experiment methodology or in the SoftMatch algorithm should be explored. The perturbation methods used in the ScanMatch and MultiMatch experiment only change the spatial offset from its twin, not the sequential order. This perturbation may describe noise introduced by machine alignment drift but does not accurately simulate the difference in perception between two different participants in a free-viewing experiment. This may describe why SoftMatch performed better at real scanpath comparison while being poor at matching simulated short-length random scanpaths. A clinical exploration of the role short sequenced analysis techniques such as SoftMatch play in measuring free-form visual search tasks can paint a complete picture of how it fits into expertise and perception. Indeed, methods such as ScanMatch and MultiMatch may provide compliments for investigating complex visual field patterns in addition to SoftMatch’s analysis of free-look similarity during a scanpath analysis.

This experiment assumed that scanpaths viewing the same stimulus would be more similar than between different stimuli, and indeed the results have shown that this is the case. However, these similarities may be driven by bottom-up saliency mechanisms, which were not explored in this research. An interesting follow-up to this research should include consistencies in strategic top-down aspects, where a task is performed, unlike the free-viewing approach used to obtain data for this research. Furthermore, an exploration of match consistency within and between participants could determine how consistently these saliency mechanisms are maintained.

Future work exploring the correlation between the length of τ values and bottom-up versus top-down processing may identify more complex fixation patterns in longer τ segments, while revealing shared bottom-up characteristics between participants with similar expertise. For example, expert radiographers may return high match scores with beginners when using shorter τ values, but may match poorly with higher τ values due to different habits with back-tracking, re-reading, etc. In this case, shorter segments returned from smaller τ values capture all the similar bottom-up search results, while longer segments from higher τ values capture top-down complex fixation patterns.

## 6. Conclusions

In this paper, we introduced a novel approach to reducing 2D scanpaths into 1D Hilbert distances to increase quantisation performance, preserve locality, and reduce complexity when integrating the temporal dimension. This approach initially provided poor results when using a small number of synthetic scanpaths in an experiment using ScanMatch and MultiMatch to test noise performance. However, it performed well in free-look scanpath testing, showing comparatively high substantive significance (>0.20) through Cohen’s effect size results, as seen in Table 1, and well defined separability seen through the magnitudes of differences in clustered heatmap quadrants in Figure 12, and through *p*-value results in Table 1.

Future work may involve investigating scanpath separability using more nuanced examples of different stimuli. For example, do people view Jackson Pollock’s Convergence painting differently to his Blue Poles one, even though they appear to be very similarly random? If these two apparently random paintings have separable scanpath patterns, what does this say about subconscious human perception? Furthermore, instead of an examination of how the same group views two different stimuli, as was done in this experiment, an examination could be made of how two different groups view the same stimulus. An expert and novice group could have their scanpaths matched when viewing the same stimulus to search for separability. For example, MRI scans which cause larger differences between scanpath matches in experts and novices may be used to address learning gaps when dealing with certain types of pathology. Most importantly, this method acknowledges the anisotropic nature of the human gaze by incorporating a combinatorial approach to scanpath matching, thereby showing results which improve upon the limitations imposed by traditional collinear methods.

## Figures and Tables

**Figure 1 sensors-22-07438-f001:**
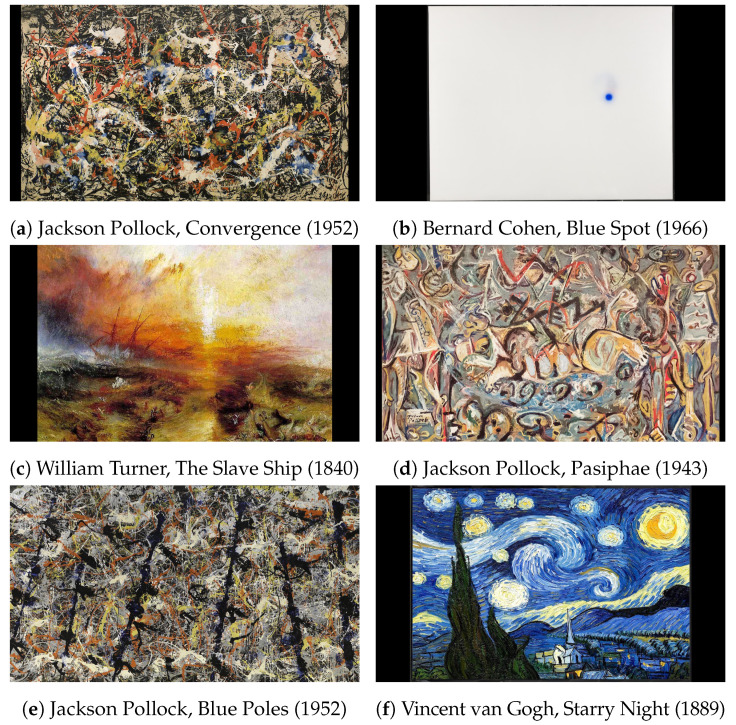
Artworks possessing various levels of abstraction and extremes in geometric complexity—e.g., Pollock’s paintings, being complex, and Cohen’s, being comparatively simple.

**Figure 2 sensors-22-07438-f002:**
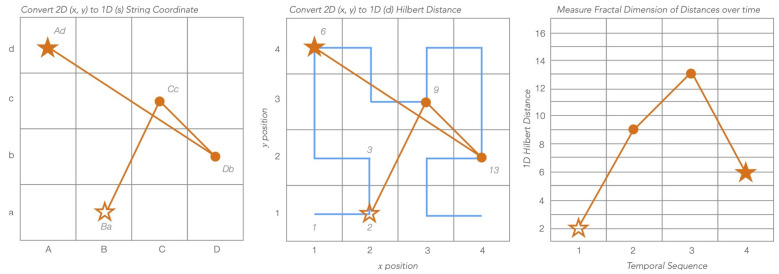
A scanpath demonstrates how a theoretical collection of four fixations could be represented over Cartesian, string, and Hilbert curve distances. All three figures represent four theoretical fixation points over a 4 by 4 unit space. The left figure illustrates how ScanMatch and MultiMatch reduce the 2D Cartesian coordinates to Ba, Cc, Db, and Ad. The middle figure illustrates a novel method for reducing the same Cartesian coordinates to Hilbert distances 2, 9, 13, and 6. A blue Hilbert curve overlay demonstrates the Hilbert curve distance’s path. The right figure shows the Hilbert curve distances to the left plotted against their temporal sequence. Start and end fixation are represented with ✩ and ★, respectively.

**Figure 3 sensors-22-07438-f003:**
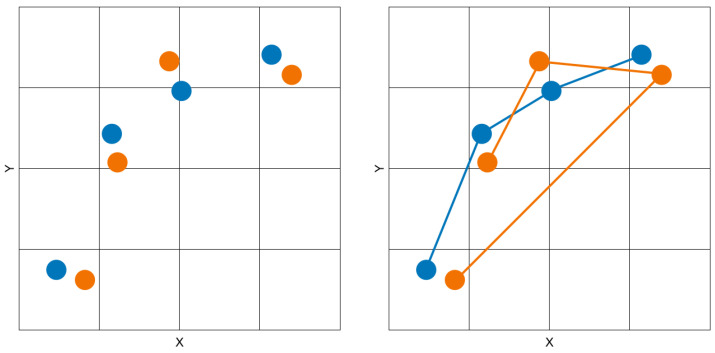
The same points expressed without a path sequence prepared for a Hausdorff distance calculation (**left**) and with a path sequence for a discrete Fréchet distance calculation (**right**).

**Figure 4 sensors-22-07438-f004:**
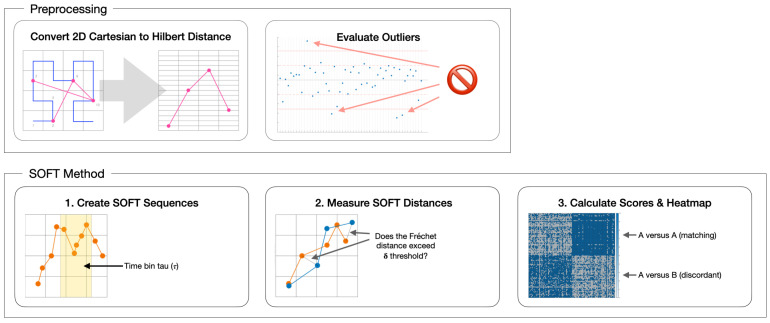
The sequences of fixations over time (SOFT)-Match scanpath comparison framework: The top block represents the preprocessing phase. The left square illustrates the conversion of the 2D pink scanpath data overlaid on a blue fractal curve path (left), with the arrow pointing to its converted data structure plotted as a pink line using x axis, time, and y axis, fractal curve position. The right square in the top block represents an outlier identification method developed by Newport et al. [37] where scanpaths exhibiting significant differences in geometric complexity are flagged for exclusion. The bottom block represents the SoftMatch method. The left square illustrates the time bin parameter τ which is used in Step 1 to establish a consistent binning size for all SoftMatch duration segments. Step 2 is where SoftMatch segment vectors, shown here as two distinct scanpath segments in blue and orange, are scored +1 if their distance is <δ. Finally, Step 3 is where a clustered heatmap reveals score comparisons between all participants. Each axis contains a box representing each of the 53 participants viewing stimuli A (positions 1 through 53) and stimuli B (positions 54 through 106). The overall 106 × 106 gridded, clustered heatmap illustrates reflective match scores in the group.

**Figure 5 sensors-22-07438-f005:**
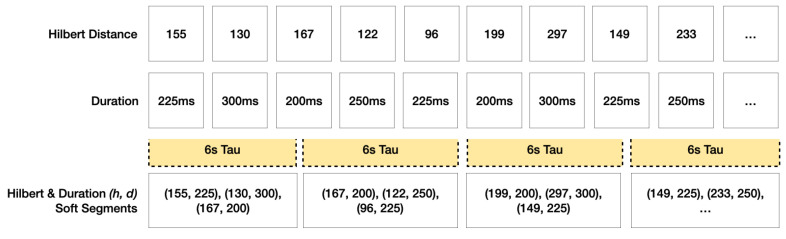
This example illustrates how fixation points, converted from (x,y) coordinates into (h) Hilbert distances, are binned using a tau τ window of 6 s. Each Soft segment, i.e., 6 s tau window, consists of a set of Hilbert and duration (h,d) pairs.

**Figure 6 sensors-22-07438-f006:**
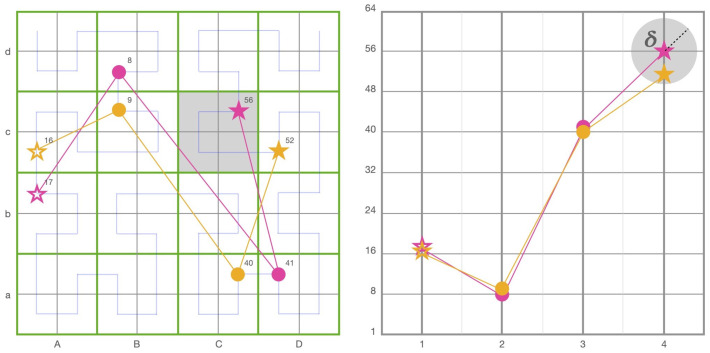
String editing quantisation methods introduce artefacts where points, depicted here as pink and orange, may be close together in Cartesian space but are far apart when quantised in the green grid. In our method, 1D Hilbert distances are quantised within the parameter δ originating from each point’s spatio-temporal position when calculating the discrete Fréchet distance, as illustrated using the fourth pink point in the diagram (**right**). Conversely, grid methods shown (**left**) quantised to enclosing green squares are prone to quantisation limitations, as shown in the diagram (**left**), where pink point Cc and orange point Dc are quantised apart even though they are close together. It should be noted that even though quantisation is reduced using this method, it is not completely removed, as demonstrated by the distance between points Bc and Cd. Start and end fixation are represented with ✩ and ★, respectively. String editing versus Hilbert distance are shown on an 8 × 8 grid quantised to 4 × 4.

**Figure 7 sensors-22-07438-f007:**
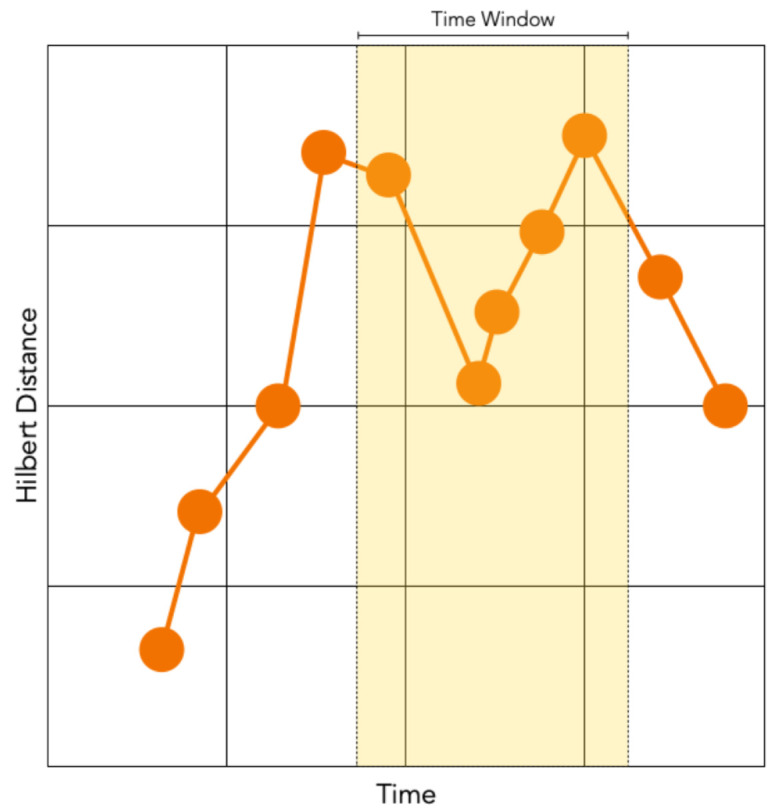
We used equally spaced time window bins to split a scanpath consisting of Hilbert distance and duration tuple values (h,d) into combinatorial segments. In cases where a fixation’s duration exceeds the time bin size, its duration is truncated; the fixation and its remaining duration are copied over to the next sequence.

**Figure 8 sensors-22-07438-f008:**
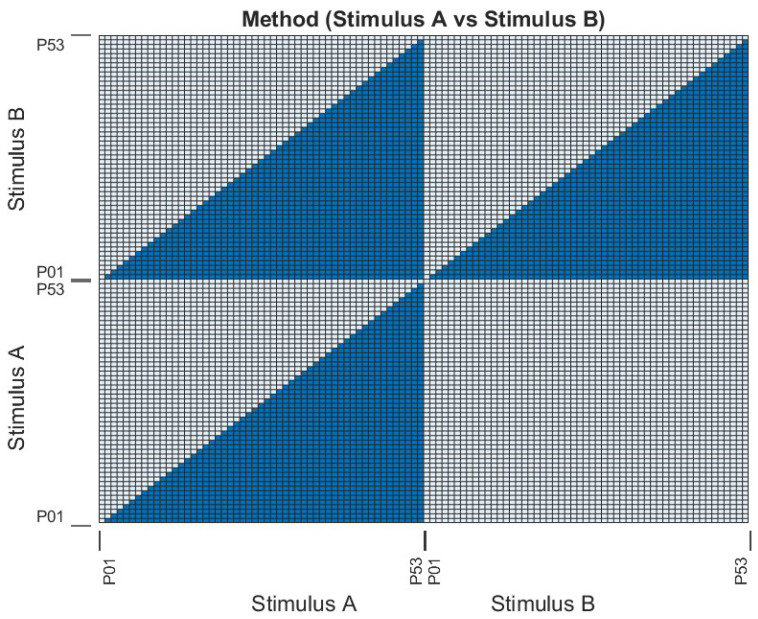
Illustration showing portions of the heatmap (solid colour) used in statistical testing. These triangular wedges omit repeating members (white) of the heatmap; e.g., match scores for (P05, P23) duplicate the match scores for (P23, P05), (P10, P10) are redundant, and all matches in Stimulus B versus Stimulus A match all those in Stimulus A versus Stimulus B.

**Figure 9 sensors-22-07438-f009:**
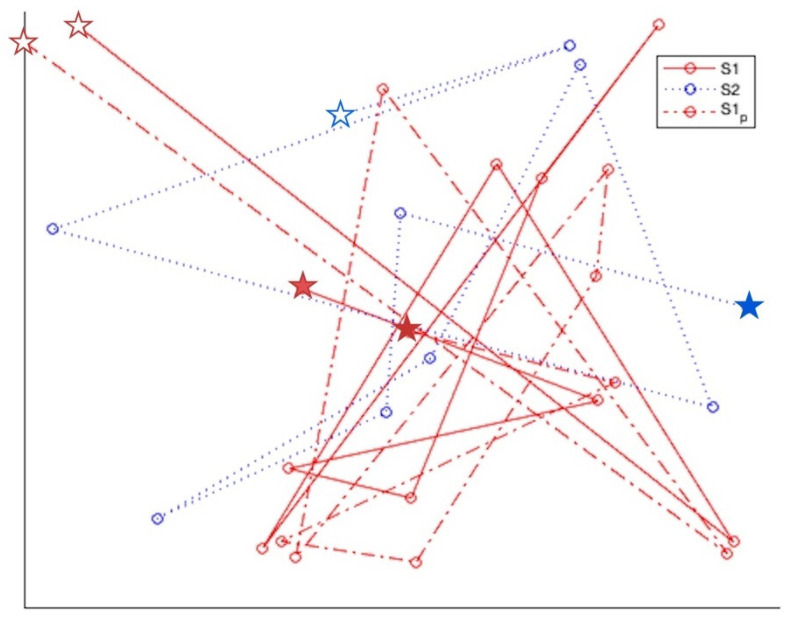
Examples of random scanpaths S1 and S2 illustrating high variability. S1p is shown as a duplicate of S1 perturbed with σ=0.1W. This experiment tests for higher similarity between S1 and S1p than with S2, given the increasing noise. Start and end fixation is represented with ✩ and ★, respectively.

**Figure 10 sensors-22-07438-f010:**
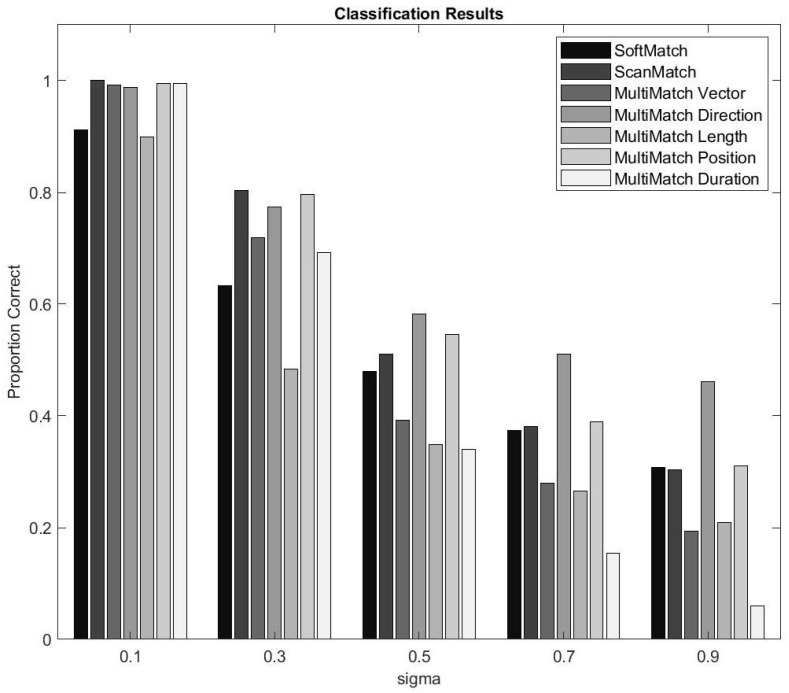
Twenty-four unique, random scanpath adversaries. S1, S2 including S1p, which is a noise perturbed version of S1. The levels of (σ) represent S1p perturbation as a measure of percentage of screen width and milliseconds between 150 and 300 ms. Each set was perturbed 50 times for a total of 6000 samples.

**Figure 11 sensors-22-07438-f011:**
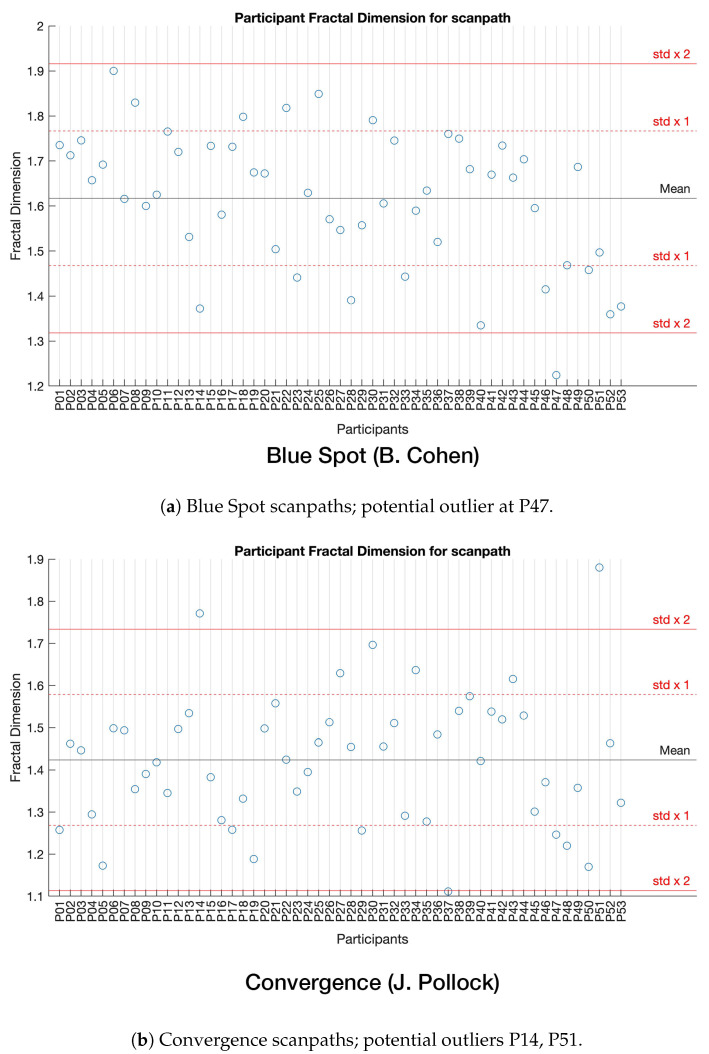
Scatter plot illustrating the geometric complexity (*y* axis) of each participant’s scanpath (*x* axis), for the purpose of outlier detection using methods from Newport et al. [37]. The black horizontal line in the approximate centre of the plot represents the mean geometric complexity. Red dotted lines represent either a 1× or a 2× standard deviation from the mean.

**Figure 12 sensors-22-07438-f012:**
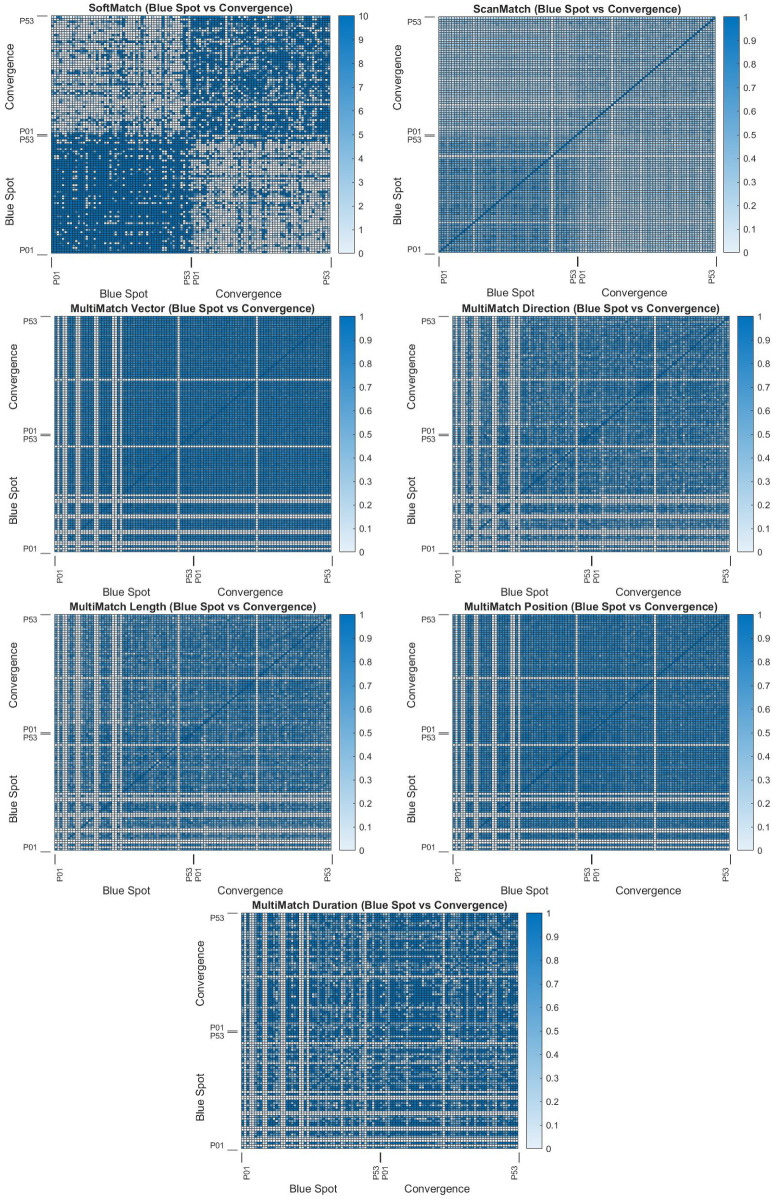
SoftMatch, ScanMatch, and MultiMatch heatmaps. Stimuli matched are Bernard Cohen’s *Blue Spot* (1966) and Jackson Pollock’s *Convergence* (1952). Darker values indicate higher matches. Complete heatmaps can be found in the Appendix A.

**Table 1 sensors-22-07438-t001:** Total results (out of 30) for each method where *p*-values are over 0.05 and Cohen’s effect size is over 0.20. Please see Appendix A for a detailed *p*-value matrix.

	*p*-Value < 0.05	Cohen’s Effect Size > 0.20
SoftMatch	24	27
ScanMatch	22	25
MultiMatch Vector	5	10
MultiMatch Direction	5	10
MultiMatch Length	4	10
MultiMatch Position	5	9
MultiMatch Duration	1	8

## Data Availability

The SoftMatch Toolbox source code written in Matlab and datasets generated during and/or analysed during the current study are available from the SoftMatch GitHub repository [44]. Experiment 1 in this research followed an identical process to that found in research by Dewhurst et al. [4]. Experiment 2 was not publicly preregistered. However, this work was mentioned as future work in a previous paper by the principal author [37].

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
