# Peer review of "SoftMatch: Comparing Scanpaths Using Combinatorial Spatio-Temporal Sequences with Fractal Curves"

_sensors, 2022, doi:10.3390/s22197438_

Round 1

Reviewer 1 Report

The authors present a new methods for comparing scanpaths to one another, introducing fractal space distance as a 1D substitute for 2D coordinates.

The topic provides an interesting problem setting and the authors' approach to solving it differs sufficiently from other methods they reference. However, I have qualms with both the chosen method, the study design, and the presentation of the material. I believe that the publication would be substantially improved by using another (publicly available) data set that would spare the authors time for collecting the data, while at the same time providing a broader base for any performed analysis. 

Major comments:

1. The amount of data, on which the method is tested, is surprisingly low - just 2 (very distinct) stimuli, ca. 50 participants free-viewing both of these. Making definitive judgements about "real-world performance" of the suggested algorithm based on this data set feels unrealistic. There are plenty of publicly available data sets at this point, including those acquired with the same eye tracker that you use in your experiments, if you wish to remain consistent in that respect. Experiments with other eye trackers would, of course, be beneficial, and the amounts of analyzed data should be increased in any case. 

2. Analysis soundness:

2.1 Statistical testing. The authors perform numerous statistical tests (from figure 12, I would estimate in the order of 1000), without any form of p-value correction for multiple testing. 

2.2 While the proposed algorithm consists of multiple novel components, and the authors claim e.g. that (l. 634) "Hilbert distances instead of traditional 2D grids is instrumental in the increased precision of quantisation", no ablation study is reported. What would the performance be if you introduced delta in (x,y,t) space instead of (h, t)? What if it was in quantized (q(x), q(y), t) space, corresponding to uniform RoI splitting? What if you used more traditional distances instead of Frechet?

2.3 In section 2.6.3 it is stated that "25 randomly picked matches" were used to optimize tau and delta values. First of all, it is not completely clear what matches are meant exactly, but if these contain real subject recordings, then this data should not be used to compute final performance metrics (e.g. should not be part of any further analysis in section 3.2. Otherwise, you unfairly favor your method in this analysis. Please clearly state what recordings were used for what purpose (algorithm parameter optimization, testing, comparison with other methods) in the paper, and make sure to keep optimization and testing/comparison data separate.

2.4 It would also be great to uniformly report results for the artificial and the real datasets, using the same metrics and plots. You can put some of this in the supplementary material, if applicable, or present the data in a different form if you wish to save space. What I specifically missed was some reporting of "proportion correct" rate for real data. 

3. Proposed method soundness: The approach that the authors follow seems overly complicated and at the same time not extremely successful, both in empirical studies and in its theoretical shortcomings.

3.1 Empirically:

3.1.1 Section 3.1, the experiment with artificial scanpaths. First, please state the observed simplicity of the test (in lines 520-523) earlier in the text. Nevertheless, while these are valid points, it is a little disturbing that the proposed supposedly better method can deal with this case worse than 4 our of 5 listed variants of MultiMatch. Temporal alignment gives a "boost" to your algorithm as well as to MultiMatch, because you will get "matching" time binning, whichever tau value you choose.

3.1.2 Section 3.2, real scanpaths. The choice of the stimuli is very limiting. First of all, the number of different stimuli is just 2. Secondly, the vastly different visual characteristics of these may not sufficiently challenge the methods under comparison, as scanpaths are bound to show very different fixation localization. 

3.1.3 This is why it is also surprising to see that the separability by the proposed method is far from perfect (e.g. derived from Figure 14, but would be nice to see a direct "percentage correct" number, too). It feels that simply comparing the saliency maps (e.g. empirically constructed averages from all observers) to the scanpaths would provide almost perfect separation. It might easily end up being a wrong assumption, but without any visualization of scanpaths from both stimuli, and especially those that were wrongly determined by the algorithm to belong to the same stimulus, it is impossible to tell whether the task is so much more difficult than it seems or the method performance is unexpectedly poor. Consider e.g. [A] and saliency map comparison as alternative to scanpath comparison, as well as for references to some publicly available data sets.

[A] Le Meur, O., Baccino, T. Methods for comparing scanpaths and saliency maps: strengths and weaknesses. Behav Res 45, 251–266 (2013). https://doi.org/10.3758/s13428-012-0226-9

3.2 Theoretically:

3.2.1 Delta in the Hilbert distance space has no interpretable mapping to Euclidean distances in the original xy space. This means that introducing delta in that space as a threshold for Frechet distance by-design lacks interpretability and "stability" - On figure 6, points in e.g. Bd and Cd quadrants with Euclidean distance of 1 will have a distance of >50 on the fractal (you mention this in the paper already, which I appreciate, but the mentions are relatively off-handed, and this does not justify designing an algorithm that creates such problems by-design in the first place).  

3.2.2 The previous point leads me to the following question: Why use any fractal mapping at all? Sure, it facilitates 2D plots of 3D data like Figure 7, but analysis so much more uninterpretable afterwards... In the subsequent analysis, you seem to mostly compute distances from the h-space coordinates. Why not use xy-space coordinates for the same distance computations directly? Maybe I am missing some point in the paper where this would make something inconsistent or impossible in the computations (not talking about visualizations at all). 

3.2.3 Tau-binning in time axis: Tau value. You both suggest for future research and try out in this paper various tau values from the 1-300ms range. From the literature, not an insignificant portion of free-viewing fixations would be pushing the upper bound of this range for duration, with distribution density peaking around 150-250ms, depending on the specifics of the papers. This to me would mean that your expected number of fixations per a tau window contains ca. 2, maybe 3, fixations on average (it those were perfectly detected by the fixation detection algorithm). Is this the case? In any case, please repot some information about fixation durations in your dataset. 

If there are indeed very few fixations in such sequences, then comparing them with complex methods (Frechet distance, Hilbert distance space, etc.) seems like an overkill... Maybe for free-viewing without expertise differentiation this is indeed irrelevant, but identifying more complex fixation patterns (e.g. in visual search - comparing candidates to search target, back-tracking in re-reading, etc.) would seem to benefit from analyzing longer fixation sequences. It would be good to at least discuss this in the paper, even if it cannot be immediately addressed/tested.

3.2.3 Tau-binning in time axis: General principle. With longer tau durations, however, which is implied by some of the plots in the paper depicting more fixations, it becomes difficult to match similar parts of the scanpaths well, because you need to be "lucky" in their temporal alignment to the beginning of the tau segment...

4. Presentation clarity. The text of the paper itself is easy to read, and individual lines of argument and justifications are sound. On the whole, it was difficult to form an impression of what exactly was done in which way, making it challenging for the reader to judge or reproduce your approach (the promised code availability is appreciated, though!). E.g. it is still not clear to me, how do the results of step 2 ("Does Frechet distance exceed delta?") for all tau-segments is aggregated into the matrix in step 3 - is it sum, average, or some other quantile? I list some minor comments below that also contribute to the overall clarity.

Minor comments:

1. Even the broad outlines of the compared methods are unclear from the paper. While they are appropriately cited, it would be appreciated if the reader unfamiliar with them would get a rough understanding of the competing approaches.

2. Figure 2 does not illustrate the distance computation

3. In 1.4, be clear about discreet vs continuous space

4. Line 138, where is 3D + time coming from? Would it not be 2D + time?

5. Line 221, the reason for "1060" is not clear.

6. In figure 4, step 3, I think the "matching" and "discordant" pointers should be switched, if I understand correctly.

7. Line 287, probably 1000Hz, not MHz. Also, why right-eye fixations were chosen?

8. Lines 354-355, about "parsing <...> attention <...> into discrete features" is unclear.

9. Line 393-394, I thinks the causality is misplaced. As far as I understand, the definition of Frechet distance requires alpha and beta to be increasing, and the inf has nothing to do with "preventing backwards computations".

Also, notations in formula (2) are a bit unclear: E.g. what does A_{i..n} mean? 

10. Caption of Figure 7 contradicts some previous statements in the paper about what happens if a fixations is on the tau-segment border: Remainder vs duplication.

11. Line 552: Statistically significant paired group differences "high separability" of the groups, at least not necessarily. 

12. Figure 11: Caption talks about tau up to 300, plot goes up too 90ms

Author Response

Reviewer 1: Major Comments

Thank you for your detailed feedback to our paper, which has led to what we hope is an improvement. We will address each comment in the same order you presented. Please see our responses immediately after each of your comments. Some of your comments are shortened for brevity. 

The amount of data, on which the method is tested, is surprisingly low - just 2 (very distinct) stimuli, ca. 50 participants free-viewing both of these. Making definitive judgements about "real-world performance" of the suggested algorithm based on this data set feels unrealistic… 

Thank you for suggesting that we include more stimuli in the analysis. Indeed, we have already captured eye tracking data that spans many different types of stimuli which we have previously included in published work. We have included four additional paintings, shown in Figure 3, all of which have varying levels of abstraction, in order to explore free viewing gaze patterns of unfamiliar stimuli. As mentioned in the paper, we do this as a proxy to high cognitive function (line 621).

2.1 Statistical testing. The authors perform numerous statistical tests (from figure 12, I would estimate in the order of 1000), without any form of p-value correction for multiple testing.

We appreciate your highlighting this weakness in the paper, and we have attempted to fix this issue with the addition of Section 3 Statistics and Testing (line 508) and the replacement of Cronbach’s alpha with Cohen’s Effect Size. Additionally, we added a table (Table 1 line 684) with direct comparisons between SoftMatch, ScanMatch, and MultiMatch p-values and Effect Sizes. A complete listing can be found in the Appendix. 

2.2 While the proposed algorithm consists of multiple novel components, and the authors claim e.g. that (l. 634) "Hilbert distances instead of traditional 2D grids is instrumental in the increased precision of quantisation", no ablation study is reported.

Thank you for highlighting this absence from the paper. Indeed, there is no formal ablation study included in the paper and we have chosen to highlight this fact on line 755. However, we do so because we would like to compare our method specifically with string edit ones used by ScanMatch and MultiMatch. 

2.3 In section 2.6.3 it is stated that "25 randomly picked matches" were used to optimize tau and delta values. …  you unfairly favor your method in this analysis. Please clearly state what recordings were used for what purpose (algorithm parameter optimization, testing, comparison with other methods) in the paper, and make sure to keep optimization and testing/comparison data separate.

Indeed, you are correct about the study unfairly favoring SoftMatch if optimisation is done only with SoftMatch and not with other methods and appreciate this insight. For this reason, we have removed optimisation from the paper, including it instead as potential for future work, as mentioned on line 312 and 502. We conducted all new tests using a SoftMatch default value of 0.90 ms for tau, obtained through the “attention window” researched by Nakayama and Motoyoshi, as mentioned on line 430.

2.4 It would also be great to uniformly report results for the artificial and the real datasets, using the same metrics and plots. You can put some of this in the supplementary material, if applicable, or present the data in a different form if you wish to save space. What I specifically missed was some reporting of "proportion correct" rate for real data.

Thank you for this insight about our lack of clarity regarding results. We hope this is addressed in Section 3 Statistics and Testing. Additionally, Figure 9 discusses the non-redundant participant samples obtained by the triangle portions of the cluster heatmap. Additionally, the “proportion correct” method of comparison is described in more detail on lines 533 and 572. Results of this are discussed using Table 1 on lines 678 through to 701.

Proposed method soundness: The approach that the authors follow seems overly complicated and at the same time not extremely successful, both in empirical studies and in its theoretical shortcomings.

Thank you for bringing our poor communication of the results to our attention. To make it more obvious that the method has merit, we have adopted a table matrix (Table 1 page 21) comparing the p-value and Effect Size measuring the magnitude of difference between concordant and discordant score comparisons, between all stimuli. 

3.1.1 Section 3.1, the experiment with artificial scanpaths. First, please state the observed simplicity of the test (in lines 520-523) earlier in the text. Nevertheless, while these are valid points, it is a little disturbing that the proposed supposedly better method can deal with this case worse than 4 our of 5 listed variants of MultiMatch. Temporal alignment gives a "boost" to your algorithm as well as to MultiMatch, because you will get "matching" time binning, whichever tau value you choose.

We appreciate your point about the method’s weakness with artificial scanpaths. However, this experiment was included to highlight how a singular approach to scanpath comparison does not necessarily mean it is the best (line 242). In this paper, we demonstrate that even though MultiMatch performed well with artificial scanpaths, it did quite poorly in the free viewing experiment. We hope it will inspire other researchers to view scanpath analysis more holistically, including separate or unique approaches to bottom up and top down analyses. 

3.1.2 Section 3.2, real scanpaths. The choice of the stimuli is very limiting. First of all, the number of different stimuli is just 2. Secondly, the vastly different visual characteristics of these may not sufficiently challenge the methods under comparison, as scanpaths are bound to show very different fixation localization.

Thank you for further elaborating on your point. We hope to have addressed this issue by using six uniquely abstract paintings seen in Figure 3.

3.1.3 This is why it is also surprising to see that the separability by the proposed method is far from perfect (e.g. derived from Figure 14, but would be nice to see a direct "percentage correct" number, too). It feels that simply comparing the saliency maps (e.g. empirically constructed averages from all observers) to the scanpaths would provide almost perfect separation.

Thank you for this suggestion. We have chosen to measure the magnitude of difference in scores between concordant and discordant matches to see if small segments of perception are shared among participants free viewing unfamiliar stimuli. We do this as a proxy for high cognitive function. We could use other methods if we simply want to match stimuli to their scanpaths. However, this would not reveal anything interesting about how the participants are engaging with bottom up processes. For this reason, we are maintaining this approach. 

3.2.1 Delta in the Hilbert distance space has no interpretable mapping to Euclidean distances in the original xy space. This means that introducing delta in that space as a threshold for Frechet distance by-design lacks interpretability and "stability" - On figure 6, points in e.g. Bd and Cd quadrants with Euclidean distance of 1 will have a distance of >50 on the fractal (you mention this in the paper already, which I appreciate, but the mentions are relatively off-handed, and this does not justify designing an algorithm that creates such problems by-design in the first place).

Thank you for pointing out this misunderstanding. Hilbert distance bends do have some limitations in small spaces. However, stability increases as resolution increases (mentioned on line 184) and due to locality preservation, increases or decreases in resolution do not effect quantisation as it does in string edit methods, making Hilbert distance dimensionality reduction more robust than the alternatives presented in this paper. The addition of a time dimension is not intended to replace or mimic Euclidean space; but rather be presented as vertices for discrete Frechet distance measurements. The experiments presented in the paper demonstrate that their application has been successful. 

3.2.2 The previous point leads me to the following question: Why use any fractal mapping at all? Sure, it facilitates 2D plots of 3D data like Figure 7, but analysis so much more uninterpretable afterwards... In the subsequent analysis, you seem to mostly compute distances from the h-space coordinates. Why not use xy-space coordinates for the same distance computations directly? Maybe I am missing some point in the paper where this would make something inconsistent or impossible in the computations (not talking about visualizations at all).

Thank you for the opportunity to clarify why Hilbert distance measurements are useful. Firstly, an interesting extension of your question would be “Why don’t ScanMatch or MultiMatch just use (x,y) space?” Indeed, the question of why we reduce dimensionality before matching anything would require a lengthy explanation, briefly explored on line 124. However, if we assume that dimensionality reduction is an important part of matching things, as the researchers of ScanMatch and MultiMatch have determined, then Hilbert distances make a better method of achieving this than string substitution, as explained on line 177 through to 192.

3.2.3 Tau-binning in time axis: Tau value. You both suggest for future research and try out in this paper various tau values from the 1-300ms range. From the literature, not an insignificant portion of free-viewing fixations would be pushing the upper bound of this range for duration, with distribution density peaking around 150-250ms, depending on the specifics of the papers…

Thank you for this question and insight. We have removed the Optimisation section and replaced optimised tau window values with a singular default value of 0.90 ms for tau, obtained through the “attention window” researched by Nakayama and Motoyoshi, as mentioned on line 430.

If there are indeed very few fixations in such sequences, then comparing them with complex methods (Frechet distance, Hilbert distance space, etc.) seems like an overkill... Maybe for free-viewing without expertise differentiation this is indeed irrelevant, but identifying more complex fixation patterns (e.g. in visual search - comparing candidates to search target, back-tracking in re-reading, etc.) would seem to benefit from analyzing longer fixation sequences. It would be good to at least discuss this in the paper, even if it cannot be immediately addressed/tested.

Thank you for this excellent insight. We have included additional notes in our discussion about the differentiation between short segments used in bottom up processing, which is well suited for SoftMatch, with longer and more complex fixation patterns which may be better suited for analysis by MultiMatch. Longer tau windows may pick up these patterns but it is outside the scope and experimental parameters set in this paper. This discussion can be found on lines 787 to 792 and 800 to 808.

3.2.3 Tau-binning in time axis: General principle. With longer tau durations, however, which is implied by some of the plots in the paper depicting more fixations, it becomes difficult to match similar parts of the scanpaths well, because you need to be "lucky" in their temporal alignment to the beginning of the tau segment…

Thank you for bringing up the misunderstanding that segments are matching due to luck. Indeed, the p-value and Effect Size results of the six paintings verify that this is not due to luck, but is a result of finding shared small segments of bottom up visual percepts.The small tau values present shorter fixation pattern segments, making this more representative of bottom up behaviors in visual scanning. Since these habits are shared in human perception, there is less of a requirement to be lucky during these matches. This would be more the case for top-down comparisons where more complex fixation patterns require matching, which is out of the scope of this paper. However, exploring how longer tau windows can capture more complex visual gaze patterns would be interesting future work. 

Presentation clarity. The text of the paper itself is easy to read, and individual lines of argument and justifications are sound. On the whole, it was difficult to form an impression of what exactly was done in which way, making it challenging for the reader to judge or reproduce your approach (the promised code availability is appreciated, though!). E.g. it is still not clear to me, how do the results of step 2 ("Does Frechet distance exceed delta?") for all tau-segments is aggregated into the matrix in step 3 - is it sum, average, or some other quantile? I list some minor comments below that also contribute to the overall clarity.

Thank you for this comment about the paper’s clarity. We have attempted to clean up the clarity using your points as a guide. Specifically, line 323 to 327 and the explanation in Figure 4 has been clarified to explain in detail how match scores are summed. 

Reviewer 1: Minor Comments

Even the broad outlines of the compared methods are unclear from the paper. While they are appropriately cited, it would be appreciated if the reader unfamiliar with them would get a rough understanding of the competing approaches.

Thank you for the suggestion to clarify the compared methods. We have attempted to be more detailed in our explanation of competing approaches on lines 85 to 114 under Section 1.1 String Edit Methods. 

Figure 2 does not illustrate the distance computation

Thank you for pointing out this lack of clarity. We have updated the verbiage to specify that these are points prepared for the two calculations on page 5, label for Figure 2. 

In 1.4, be clear about discreet vs continuous space

Thank you for the prompt to be more clear about discrete versus continuous space. We have added an explanation on line 177. 

Line 138, where is 3D + time coming from? Would it not be 2D + time?

Thank you for pointing out this error, it has been corrected on line 182.

Line 221, the reason for "1060" is not clear.

Thank you for pointing this out. It does not add insight and only creates confusion, so it has been removed. 

In figure 4, step 3, I think the "matching" and "discordant" pointers should be switched, if I understand correctly.

Thank you for pointing out his confusion. The figure is correct, as higher scores mean more matchability, as indicated by a darker colour. We have attempted to make this more clear through the description on lines 668 to 683.

Line 287, probably 1000Hz, not MHz. Also, why right-eye fixations were chosen?

Thank you for pointing out this error, and asking this important question about using the right eye. We have included an explanation about how “sometimes one eye is better than two” on line 339 to 334.

Lines 354-355, about "parsing <...> attention <...> into discrete features" is unclear.

We appreciate your pointing out this lack of clarity. We have re-worded it on lines 413 to 414. 

Line 393-394, I thinks the causality is misplaced. As far as I understand, the definition of Frechet distance requires alpha and beta to be increasing, and the inf has nothing to do with "preventing backwards computations".

Thank you for finding this misplaced causality. We have reworded it be more clear on line 452 to 454.

Also, notations in formula (2) are a bit unclear: E.g. what does A_{i..n} mean?

We thank you and agree that it is unclear. We have changed Equation 2 to include “Let i…n be fixation sequences of parent scanpath”.

Caption of Figure 7 contradicts some previous statements in the paper about what happens if a fixations is on the tau-segment border: Remainder vs duplication.

Thank you for pointing out this confusing verbiage. We have changed it to be more clear in the caption for Figure 7 on page 14.

Line 552: Statistically significant paired group differences "high separability" of the groups, at least not necessarily.

Thank you for this comment. However, we have removed Optimisation from this paper and intend to pursue it in future work. 

Figure 11: Caption talks about tau up to 300, plot goes up too 90ms

Thank you again for this comment. As we mentioned above, we have removed Optimisation from this paper.

Reviewer 2 Report

The manuscript is very nicely formulated and presents a novel idea for scanpath analysis that seems to be going in a very promising direction. The authors detail a method for comparing the similarity of scanpath using fractal curves and new (to scanpaths) distance metric. The authors point out weakness in the traditional approaches and use that as motivation for their new method. Overall, I found the paper easy to read, especially given the complexity of the content and the authors make a strong claim to their method and use appropriate and rigorous evaluations of their method compared to the current state of the art. They also provide access to the data and toolbox used, which promotes reproducability in the community.  For an eye tracking analysis paper, they provide a fitting contribution to the eye tracking research community, I can only hope that eye tracking researchers will be able to find this paper easily, if accepted.

As positive as this review sounds, I do have major concerns that should be addressed:

1. You mention both MultiMatch and ScanMatch as states of the art, but you only compare your method to MultiMatch. ScanMatch should also still have a Matlab implementation and you could choose a highly generic Match,Mismatch,and Gap penalty and relatively quickly compare your method to ScanMatch. I feel this would be a really nice addition to section 3.2. Even though the string edit approaches have their limitations, they can be annoyingly good at determining similarity. I feel like this is why most research wind up using it because it often tends to work well enough.

2. [Moreso minor comment] Because of the just mentioned statement that scanmatch (and other string alignment) tend to work sufficiently well for most scanpath analyses. It is always good to really point out in your introduction as well as your discussion, why should future researchers move away from this traditional approach and use your approach. Already, you make some convincing points about linearity as well as spatial scaling, but researchers may need a bit more convincing

3. This is somewhat major, your participants perform a free viewing task on 2 images. Why did you choose this experimental paradigm over a more task-based paradigm? A more appropriate method could be free-viewing scanpaths compared to task-driven scanpaths (e.g. locate a specific feature or object in the image). Your stimuli are 2 very different abstract art works and some scanpath literature points out that scanpaths related to different images are easier to differentiate compared to scanpaths related the same image but give a different task to inspect the object. See works by [A. Haji-Abolhassani and J. J. Clark, “An inverse yarbus process: Predicting observers’ task from eye movement patterns,” Vision Research, vol. 103, pp. 127–142, 2014.], [M. R. Greene, T. Liu, and J. M. Wolfe, “Reconsidering yarbus: A failure
to predict observers’ task from eye movement patterns,” Vision Research,
vol. 62, pp. 1–8, 2012.], [M. Lipps and J. B. Pelz, “Yarbus revisited: Task-dependent oculomotor behavior,” Journal of Vision, vol. 4, no. 8, p. 115, 2004.], [A. Borji and L. Itti, “Defending yarbus: Eye movements reveal observers’
task,” Journal of Vision, vol. 14, no. 3, pp. 1–22, 2014.] for some more info in scanpath analysis (and some of it's unsolved problems...).

Minor concerns

1. Redundancy issues. some sentences could be removed as they are redundant and mentioned already earlier (e.g. whole second paragraph in section 2.2) 

2. too many commas first sentence of abstract, makes the sentence harder to read

3. Some other cool papers have started looking at fractals to assess the complexity of scanpaths, you mention Gandomkar, but [Alamudun, F., Yoon, H. J., Hudson, K. B., Morin‐Ducote, G., Hammond, T., & Tourassi, G. D. (2017). Fractal analysis of visual search activity for mass detection during mammographic screening. Medical physics, 44(3), 832-846.] also looked into fractals. I can't think of any other papers off the top of my head, but I thought it may be interesting

4. figures 1,6,8: sometimes it's nice to indicate the start of the scanpath with another symbol (e.g. star, or unfilled circle). it can help the readability of the figure without having to rely on the actual text.  

5. another paper you may like [R.-F. Day, “Examining the validity of the needleman–wunsch algorithm in identifying decision strategy with eye-movement data,” Decision Support Systems, vol. 49, no. 4, pp. 396–403, 2010.]

6. Regarding section 2.6. I am just curious how different your distance metric value would be if you replaced the Hilbert with a simpler distance metric (euclidean or hamming). I like your choice for the Hilbert and you support it well, but would be slightly interesting to see how the simpler methods compare to it. Do you have any comments on this?

7. use "that" instead of "which" at lines 423 and 450   

I feel this paper makes a promising contribution to the eye tracking research community and if the authors address my concerns, I would promote acceptance of this manuscript

Author Response

Reviewer 2: Major Comments

Thank you for your detailed feedback to our paper, which has led to what we hope is a clarifying rewrite of the paper. We will address each comment in the same order you presented. Please see our responses immediately after each of your comments. Some of your comments are shortened for brevity. 

You mention both MultiMatch and ScanMatch as states of the art, but you only compare your method to MultiMatch. ScanMatch should also still have a Matlab implementation and you could choose a highly generic Match,Mismatch,and Gap penalty and relatively quickly compare your method to ScanMatch. 

Thank you for this great suggestion. We have included ScanMatch in both the Artificial and Real World tests. Indeed, it did do well in both tests, but fell slightly short in the real world tests. This is explored in Section 4.1 and 4.2 as well as in the discussion. A table outlining results can be found on page 21 in Table 1. 

[Moreso minor comment] Because of the just mentioned statement that scanmatch (and other string alignment) tend to work sufficiently well for most scanpath analyses. It is always good to really point out in your introduction as well as your discussion, why should future researchers move away from this traditional approach and use your approach. Already, you make some convincing points about linearity as well as spatial scaling, but researchers may need a bit more convincing

We appreciate the insight into how we can better promote this method. Indeed, our new table on page 21 indicating higher p-values and Effect Sizes than all other methods is a first step. Additionally, on line 787 we discuss how SoftMatch is well suited for small segments typically representative of bottom up search processing, but could be well complimented by methods like MultiMatch or ScanMatch for more complex visual pattern matching. 

This is somewhat major, your participants perform a free viewing task on 2 images. Why did you choose this experimental paradigm over a more task-based paradigm? A more appropriate method could be free-viewing scanpaths compared to task-driven scanpaths

Thank you for the insight and follow up question. As discussed on line 57 to 65, we consider this free viewing experiment to be a proxy for high cognitive function. In previous research (10.1371/journal.pone.0260717), we have found that unfamiliar stimuli will lead to greater time spent in bottom up processing. The artworks, being mostly abstract and unfamiliar, will provoke a lot of searching from the participants, making task based scanpath matching very challenging for methods used to match patterns. With SoftMatch, we would like to introduce a method well suited to such experiments, to explore how bottom up processes compare between participants. This can be a useful task when researching expertise and perception. 

Reviewer 2: Minor Comments

Redundancy issues. some sentences could be removed as they are redundant and mentioned already earlier (e.g. whole second paragraph in section 2.2)

Thank you for this comment. We have removed the paragraph. 

too many commas first sentence of abstract, makes the sentence harder to read

We appreciate the feedback and have made the abstract easier to read. 

Some other cool papers have started looking at fractals to assess the complexity of scanpaths…

Thank you for this suggestion. We have future work planned using geometric complexity and will include these papers in that work. 

figures 1,6,8: sometimes it's nice to indicate the start of the scanpath with another symbol (e.g. star, or unfilled circle). it can help the readability of the figure without having to rely on the actual text.

Thank you for this great suggestion. We have incorporated star symbols in our illustrations. 

another paper you may like [R.-F. Day, “Examining the validity of the needleman–wunsch algorithm in identifying decision strategy with eye-movement data,” Decision Support Systems, vol. 49, no. 4, pp. 396–403, 2010.]

Thank you for this additional paper for consideration. We have incorporated this into line 39 to 47.

Regarding section 2.6. I am just curious how different your distance metric value would be if you replaced the Hilbert with a simpler distance metric (euclidean or hamming). I like your choice for the Hilbert and you support it well, but would be slightly interesting to see how the simpler methods compare to it. Do you have any comments on this?

We appreciate this question, and did an ablation study during the exploration of methods used in this paper. However, we do not formally include them. Though, mentioning the missing study should be done in the Discussion section, which we included on line 755 to 761. 

use "that" instead of "which" at lines 423 and 450

Thank you, this has been fixed on line 483, with the other removed due to editing.

Round 2

Reviewer 1 Report

Thank you for the revision of the article, as well as the more precise explanations about the nature and the target of the experiments. The bottom-up nature and the additional details about the literature context do make it more clear to me. I believe that the paper is acceptable almost in the present form - with addressing the following small remarks below. Great work on the revision! 

For the clarity of presentation, I would ask to rephrase the eq (2) or maybe its accompanying text again - it is unclear whether all possible i/j pairs of fixation subsequences (the length of tau?) in A and B are compared, or only the ones with the same index. Moreover, the notation for [D] looks a little strange, as it is then re-used in the curly brackets but after the equality sing. If this formula requires several steps, please be more verbose to correctly explain it. Also, are you talking about fixations (as in the eye movement that spans a certain amount of time) or gaze samples here?

I have another small issue with Figure 11, since you claim that no outliers were present in the updated analysis, I am confused by the sub-captions "<...> outliers P<...>". 

One last note: In the discussion, you offhandedly mention machine learning and neural networks, without a clear tie in. Optimization of the 2 method parameters is a direct follow-up on the content of the paper, but ML and even specifically NNs are a much longer reach without additional explanation. I do appreciate the other changes in the discussion, however! 

Minor typo-like remaining comments:

* At the end of 1.1, "an examination of the human gaze should be investigated." - typo/misformulation?

* Beginning of section 2, it should be "captures" now

* Figure 4 caption might need to be adapted, since we are now talking about more than 2 stiumli, so the "both stimuli" is not entirely understandable.

* I would suggest re-ordering the bars in Figure 10 to have your algorithm either first or last for better readability 

* "we used an existing dataset" in 4.2: Maybe just reference the section in your paper about what the data contains (with "existing dataset" somehow a reference would be expected, it seems)

* In section 5, I assume that "substantive (> 0.02)" should have been "> 0.2"

Author Response

Thank you for your kind words and constructive feedback. Indeed, we have put in a lot of work in the revision and agree that the changes have resulted in a stronger manuscript. We will now attempt to address the current comments in this latest version.

Reviewer 1: Comments

For the clarity of presentation, I would ask to rephrase the eq (2) or maybe its accompanying text again - it is unclear whether all possible i/j pairs of fixation subsequences (the length of tau?) in A and B are compared, or only the ones with the same index. Moreover, the notation for [D] looks a little strange…

Thank you pointing out this poorly formed equation. The i/j pair measurements are not limited to the same index, and Equation 2 on page 11 has been updated to reflect this. Additionally, the [D] notation is removed.

I have another small issue with Figure 11, since you claim that no outliers were present in the updated analysis, I am confused by the sub-captions "<...> outliers P<...>".

Thank you for pointing this out to us. We have added the word “potential” to communicate that the points were not conclusively defined as outliers. Figure 11 is on page 18.

One last note: In the discussion, you offhandedly mention machine learning and neural networks, without a clear tie in. Optimization of the 2 method parameters is a direct follow-up on the content of the paper, but ML and even specifically NNs are a much longer reach without additional explanation. I do appreciate the other changes in the discussion, however! 

We agree that there should be some detail into how this may be performed. This addition can be seen on line 662 to 665.

At the end of 1.1, "an examination of the human gaze should be investigated." - typo/misformulation?

Thank you for pointing out this poorly worded sentence. We have changed it on line 111 to 113.

Beginning of section 2, it should be "captures" now.

Thank you for catching this, it has been corrected on line 286.

Figure 4 caption might need to be adapted, since we are now talking about more than 2 stiumli, so the "both stimuli" is not entirely understandable.

Thank you pointing this out. We agree and have corrected it on page 4, Figure 4. 

I would suggest re-ordering the bars in Figure 10 to have your algorithm either first or last for better readability 

Thank you for this suggestion, we have made this change and it can be found on page 16, Figure 10. 

"we used an existing dataset" in 4.2: Maybe just reference the section in your paper about what the data contains (with "existing dataset" somehow a reference would be expected, it seems)

Thank you for this suggestion, we have included a reference on line 575.

In section 5, I assume that "substantive (> 0.02)" should have been "> 0.2”

Thank you for finding this error, we have corrected it on line 652.